

# Self-congruent point in critical matrix product states: An effective field theory for finite-entanglement scaling

Jan T. Schneider[1][*], Atsushi Ueda[2,3], Yifan Liu[3], Andreas M. Läuchli[4,5], Masaki Oshikawa[3,6,7] and Luca Tagliacozzo[1][†]

**1** Institute of Fundamental Physics IFF-CSIC, Calle Serrano 113b, 28006 Madrid, Spain
**2** Department of Physics and Astronomy, University of Ghent, 9000 Ghent, Belgium
**3** Institute for Solid State Physics, University of Tokyo, Kashiwa 277-8581, Japan
**4** Laboratory for Theoretical and Computational Physics, PSI Center for Scientific Computing, Theory and Data, 5232 Villigen, Switzerland
**5** Institute of Physics, École Polytechnique Fédérale de Lausanne (EPFL), 1015 Lausanne, Switzerland
**6** Kavli Institute for the Physics and Mathematics of the Universe (WPI), University of Tokyo, Kashiwa 277-8583, Japan
**7** Trans-scale Quantum Science Institute, University of Tokyo, Bunkyo-ku, Tokyo 113-0033, Japan

* jan.schneider@iff.csic.es , † luca.tagliacozzo@iff.csic.es

## Abstract

We set up an effective field theory formulation for the renormalization flow of matrix product states (MPS) with finite bond dimension, focusing on systems exhibiting finite-entanglement scaling close to a conformally invariant critical fixed point. We show that the finite MPS bond dimension $\chi$ is equivalent to introducing a perturbation by a relevant operator to the fixed-point Hamiltonian. The fingerprint of this mechanism is encoded in the $\chi$-independent universal transfer matrix's gap ratios, which are distinct from those predicted by the unperturbed Conformal Field Theory (CFT). This phenomenon defines a renormalization group *self-congruent point*, where the relevant coupling constant ceases to flow due to a balance of two effects; When increasing $\chi$, the infrared scale, set by the correlation length $\xi(\chi)$, increases, while the strength of the perturbation at the lattice scale decreases. The presence of a *self-congruent point* does not alter the validity of the finite-entanglement scaling hypothesis, since the self-congruent point is located at a finite distance from the critical fixed point, well inside the scaling regime of the CFT. We corroborate this framework with numerical evidence from the exact solution of the Ising model and density matrix renormalization group (DMRG) simulations of an effective lattice model.

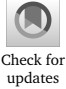

## Contents


# 1 Introduction

Quantum many-body systems continue to elude our understanding, especially in the regime of strong correlations, where intriguing collective behavior such as superconductivity emerges. This challenge is traditionally attributed to the curse of dimensionality, where the Hilbert space dimension of a quantum many-body system grows exponentially with the number of its constituents. Tensor networks offer a variational approach to studying quantum many-body systems by encoding their properties in constructing a network of elementary tensors [1]. This method mitigates the curse of dimensionality by reducing the complexity of quantum many-body systems to the complexity of contracting the tensor network. The arguably most successful example of tensor networks are matrix product states (MPS), which form the class of a variational ansatz and the foundation of the density matrix renormalization group (DMRG) [2,3].

MPS was initially conceived to represent a class of one-dimensional (1D) wave functions, which are ground states of a special class of 1D Hamiltonians [4–7]. Through a quantum information perspective on MPS, their nature was later clarified [8]; MPS are 1D wave functions with limited entanglement, reflected in the finite size of their elementary tensor. Such a class of wave functions has universal importance beyond their original conception. For example, MPS approximates the ground states of a 1D generic gapped Hamiltonian with short-range interactions well, with the elementary tensor's size fixed. These ground states obey the "area law" of entanglement, i.e., the entanglement of a region scales proportionally to the surface of that region rather than its volume [9]. In one dimension, two points form the surface of a

connected region. The area law hence implies that entanglement does not increase with the size of the block but is bounded by a constant value, which determines the necessary bond dimension of the elementary tensors. Consequently, as long as the entanglement is limited, the tensor network can describe correlated quantum states in the thermodynamic limit by concatenating infinitely many times the same few tensors constituting the unit cell. This was already discussed in the pioneering Affleck–Kennedy–Lieb–Tasaki (AKLT) papers [4,5] in one and higher dimensions, although these states were regarded as very special states at that time.

The use of tensor networks as a basis of systematic numerical approximations to general quantum systems was discussed for one dimension in Ref. [10], and for higher dimensions in Refs. [11, 12]. Considering the thermodynamic limit is very appealing when using MPS to extract the universal properties of critical, i.e., gapless systems. Such systems constitute an exception to the area law as their correlation length diverges, leading to a correction in the entanglement entropy proportional to the logarithm of the length of the block. In Refs. [13,14], it was shown that studying critical systems using MPS as a variational ansatz reveals a new form of scaling, where the correlation length is dictated by the bond dimension of the tensors rather than the system size that is taken to be infinity from the beginning. This scaling was initially formulated phenomenologically by appealing to the scaling hypothesis. It is typically referred to as finite-entanglement scaling. Using Conformal Field Theories (CFTs), theoretical insights into the emergence of such scaling were reported in Refs. [15, 16]. This idea later has been applied to several fields, including two-dimensional quantum systems [17–23]. Phenomenological studies have extended to the similar finite-correlation scaling of 2D infinite PEPS wavefunctions [24–26] and recently to out-of-equilibrium protocols [27, 28], but the puzzle of the theoretical origin of this scaling remains unsolved [29, 30]. Early discussions suggested that the finite bond dimension could be interpreted within the renormalization group (RG) framework. Studying a conformally invariant critical point with an MPS of finite bond dimension should correspond to adding a relevant perturbation to the CFT. In contrast, the perturbation depends on the bond dimension, akin to interpreting finite size as a relevant perturbation in the RG sense [24]. In recent years, such an interpretation has been confirmed quantitatively in various settings [22, 23, 31]. Lastly, the MPS transfer matrix (TM) spectrum, in the finite entanglement regime, has been shown to coincide with the infinite DMRG low-energy effective Hamiltonian spectrum.

It was previously suggested that this TM spectrum could correspond to the finite size spectrum of a CFT defined on an appropriate geometry, either a chain with periodic boundary conditions or a finite chain with the proper boundary conditions, and thus could be predicted by using an appropriate boundary conformal field theory (BCFT) [16]. Attempts to match the known BCFT spectra to the TM spectra have failed. Later, it has been conjectured that rather than a uniform chain, the spectrum should correspond to that of a chain with an impurity at the center [29,30], originating from the replicated structure of the MPS transfer matrix. Nevertheless, the attempts to fix the strength of the impurity on the top of known BCFT spectra [32] to reproduce the TM spectra had been unsuccessful [22]. Interpreting finite-entanglement scaling as the presence of a relevant perturbation led to the conjecture that this transfer matrix spectrum might be related to a new RG fixed point, similar to the $E8$ fixed point identified by Zamolodchikov when perturbing the Ising model with a magnetic field [33]. Initial attempts to match the MPS spectrum with that unveiled by Zamolodchikov and Fonseca [34] did not succeed [22] even though these studies allowed, for example to realize that the transfer matrix spectra in space and time are related by re-scaling with the velocity of sound (for later applications of this observation see [35]).

In this paper, we reconsider this picture by performing a careful RG analysis of a critical system containing both an ultraviolet and an infrared cutoff (an effective field theory) whose microscopic Hamiltonian is deformed via a relevant perturbation. The infrared cut-off mimics

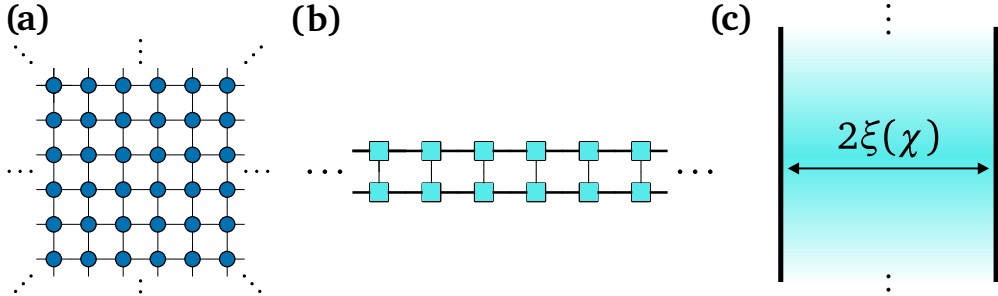

Figure 1: Sketch of an effective field theory for finite-entanglement scaling: **(a)** A Lorentz-invariant CFT on the infinite plane can be regularized on a lattice giving rise to a 2D tensor network, encoding the partition function of a 2D classical model or equivalently the discretized path integral of a quantum chain. **(b)** The latter ground states with infinite system size can be numerically approximated using MPS algorithms with finite bond dimension $\chi$. The norm of the MPS thus represents the partition function. **(c)** In this work, we unveil that such a finite-$\chi$ MPS can be thought of as corresponding to a field theory confined to length $2\xi(\chi)$ with boundaries, hence giving rise to a strip geometry for the 2D partition function. Moreover, the field theory is no longer the original CFT; rather, the CFT is perturbed with a relevant operator that encodes the errors induced by the finite bond dimension of the MPS. Such an error causes the strength of the perturbation to decrease as we increase the bond dimension. At the same time, the width of the strip and the IR-cutoff increase with increasing the bond dimension. The two effects thus compensate, and give rise to a *self-congruent point*. In this work, we discuss the effective Hamiltonian encoding of such ingredients. We also explain how the symmetry properties of the elementary MPS tensors dictate which relevant operator is dominant and which boundary conditions are thus imposed at the boundaries of the effective strip.

the presence of the finite correlation length induced by the finite bond dimension of the MPS. The strength of the perturbation at the lattice scale (the field theory UV cutoff) decreases as we increase the infrared cutoff. The combined effect gives rise to a *self-congruent point*, that can only be distinguished by the real CFT fixed point by measuring properties of the theory that probe scales comparable to the infra-red cutoff, such as the MPS transfer-matrix spectrum. As a result, the presence of such a self-congruent point does not affect the validity of the finite-entanglement scaling hypothesis, given that the RG flow from the lattice scale to the infra-red cutoff is the one predicted by the CFT. The RG flow of the effective theory of the MPS just halts exactly at the self-congruent point which lies in the scaling regime of the CFT but remains at finite distance from the CFT fixed point. We quantitatively confirm that a relevant perturbation significantly affects the TM spectrum. We characterize this picture using the Ising model as the paradigmatic example.

Figure 1 visualizes the take-home message of this work. While the intended goal is to simulate the Lorentz-invariant CFT described by a path integral or a partition function on the infinite plane, we can only approximate those, e.g., using tensor network algorithms. Such approximation introduces a relevant perturbation in the RG sense. We reveal that the low-energy spectrum of the MPS transfer matrix corresponds to the low-energy spectrum of an appropriately perturbed BCFT. The BCFT is defined on an infinite strip whose width provides the IR cut-off. We can thus interpret the MPS ground state as the exact ground state of a finite chain whose boundary conditions are dictated by the symmetry properties of the elementary tensors in the MPS. Free boundary conditions on both sides of the strip are necessary to describe the results obtained using MPS made by explicitly symmetric tensors, as well as fixed boundary

conditions for non-symmetric tensors. Furthermore, the BCFT is perturbed by a relevant perturbation whose strength at the UV cutoff decreases as the strip width increases. This affects the ratio of the gaps of the CFT transfer matrix along the strip, which we identify with the MPS transfer matrix.

This paper is organized as follows: In Section 2, we recapitulate previously established results on finite-entanglement scaling and analytical knowledge from Conformal Field Theory. In particular, we repeat the calculations of the transfer matrix spectrum of the critical Ising model in the thermodynamic limit up to a large bond dimension and extract the scale-invariant converged values.

In Section 3, we reconsider the argument put forward in Refs. [16, 22, 23, 31], that simulating critical systems with MPS with finite bond dimension $\chi$ can be described in terms of an effective model where the critical Hamiltonian is perturbed by a relevant perturbation driving the system away from the critical point. In particular, we show that as the system size increases, keeping the MPS bond dimension constant, the expectation value of a local observable behaves as if we were perturbing the fixed point Hamiltonian with the most relevant perturbation. The Ising model is invariant under $\mathbb{Z}_2$ symmetry and all scaling operators are divided into either $\mathbb{Z}_2$ charge sector—even or odd. The most relevant operator is the magnetic operator $\sigma$, being in the odd sector, while in the even sector, the most and only relevant one is the thermal operator $\varepsilon$. We show that depending on using symmetric tensors or ordinary tensors, we observe perturbations by either $\varepsilon$ or $\sigma$, respectively. When we do not explicitly enforce $\mathbb{Z}_2$ symmetry in our numerics, both sectors are numerically available and the most relevant perturbation is the magnetic operator $\sigma$. When enforcing the symmetry exactly in our numerics, all odd operators are not permitted under the symmetry and the most relevant even operator is the thermal operator $\varepsilon$. We thus confirm that the onset of finite-entanglement scaling can be associated with the appearance of a relevant perturbation within the RG framework.

In Section 4, we turn back to the infinite system, and we explore the origin of the scale-invariant spectrum of the MPS transfer matrix (TM) in infinite DMRG, which differs from the CFT predicted values. First, we reiterate the puzzle at hand and reason how the TM spectrum becomes scale-invariant by reconsidering the emergence of such a new "pseudo fixed point" and identifying it as a *self-congruent point*. To understand this self-congruent point, it is crucial to understand that increasing the bond dimension reduces the relevant perturbation at the lattice scale. However, at the same time, we change the infrared scale, given by the correlation length $\xi(\chi)$, which sets the maximum scale up to which the RG flow acts. Using the RG flow equations, one can show that the two effects compensate and the running of the couplings with the renormalization scale 'freeze' at a finite distance from the CFT fixed point.

In Section 5, we confirm the above scenario through an effective lattice model which we study with the analytical Gaussian fermionic representation of the Ising model, as well as DMRG simulations. This effective lattice model is obtained by perturbing the critical Ising model with the only relevant $\mathbb{Z}_2$-invariant operator, the thermal perturbation $\varepsilon$, using the RG equation to fix the strength of the perturbation to the scale of validity of the effective model. Within this framework, we show that we obtain the desired results of the lowest energy spectral ratios to those of the TM spectrum at the self-congruent point. Given our explanation through an effective field theory, we argue that the transfer matrix spectrum is universal, which we corroborate through observing the same spectrum at a critical point in a different model (Appendix B) which is also in the Ising universality class. Additionally, we give reason as to why the relevant perturbation settles in the spontaneous symmetry broken phase as opposed to in the paramagnetic phase on the other side of the phase transition and corroborate our argument in the critical three-states Potts model (Appendix C), where we find the same to occur. Our effective model explains the convergence of the transfer matrix spectrum to values independent of the bond dimension but different from those predicted by the unperturbed

BCFT, such as the spectrum of a critical Hamiltonian on a finite ring or on a finite segment with different types of boundary conditions. Thus, our results provide a crucial step toward fully understanding the full field theoretical picture of the observed scale invariance underlying the finite-entanglement and finite-correlation length scaling observed numerically.

## 2  Short summary of the phenomenology of finite-entanglement scaling

As mentioned in the introduction, MPS and PEPS can directly describe systems in the thermodynamic limit. These algorithms can avoid the boundary effects that finite-size simulations often suffer by exploiting translation invariance. However, the exact representation of the translationally invariant states often requires infinite degrees of freedom, making it impossible to realize on finite computational resources. Instead, we introduce a rank-3 tensor $A^i_{\alpha\beta}$ represented graphically in the Penrose notation as,

$$A^i_{\alpha\beta} = \quad \alpha \overset{i}{\underset{}{\longrightarrow}} \boxed{A} \longrightarrow \beta \,, \tag{1}$$

where $\alpha$ and $\beta$ have dimension $\chi$, and $i$ is an index for the $d$ physical degrees of freedom, e.g., the $d = 2S + 1$ spin states in spin-$S$ systems. As a result, $A^i_{\alpha\beta}$ contains a finite number of elements, namely $d\chi^2$. We can now concatenate infinite copies of $A^i_{\alpha\beta}$ and obtain a translationally invariant state in the thermodynamic limit, an infinite MPS (iMPS), graphically represented as,

$$|\psi_{\text{iMPS}}\rangle \equiv \quad \cdots \longrightarrow \boxed{A} \longrightarrow \boxed{A} \longrightarrow \boxed{A} \longrightarrow \boxed{A} \longrightarrow \cdots \tag{2}$$

The correlations in such an iMPS are mediated by the MPS transfer matrix $E$, which is defined as the contraction,

$$E^{(\beta,\delta)}_{(\alpha,\gamma)} = \sum_i A^i_{\alpha,\beta} \bar{A}^i_{\gamma,\delta} = \quad \begin{matrix} \gamma \longrightarrow \boxed{\bar{A}} \longrightarrow \delta \\ \quad | \\ \alpha \longrightarrow \boxed{A} \longrightarrow \beta \end{matrix} \quad = (\alpha,\gamma) \longrightarrow \boxed{E} \longrightarrow (\beta,\delta) \,, \tag{3}$$

where $\bar{z}$ marks the complex conjugate of $z$, and we combine the two bond indices corresponding to the right and left side of transfer matrix $E$ into a super index, respectively, and thereby rendering $E$ indeed into a matrix with two indices. It is well known that MPS describe states with exponentially decaying correlation functions [36], and that the corresponding correlation length $\xi$ has a simple form in terms of the ratio of the two leading eigenvalues of $E$ (3), $\xi = -\log e_2/e_1$. It is thus useful to define the spectrum of the transfer matrix as,

$$\Delta^E_i = -\log(e_i). \tag{4}$$

Consequently, it is not surprising that the best approximations to critical states using an infinite MPS with finite bond dimension describe wave functions with exponentially decaying correlations. The correlation length of the state increases algebraically with the bond dimension of the elementary tensor $\xi(\chi) \propto \chi^\kappa$. The first observations of this fact were made in Refs. [13, 14], and we summarize the consequences of these observations here. Given a finite bond dimension $\chi$, the variational algorithms typically allow one to reach the scaling regime close to the critical point. There, by virtue of the scaling hypothesis, all universal quantities depend only on the correlation length $\xi(\chi)$. More precisely, the scaling hypothesis states that

the free energy, or equivalently the correlation length, close to a continuous phase transition point is a generalized homogeneous function [37]. For example, an operator $\hat{O}$ with leading support on a scaling field with dimension $x_O$, whose expectation value should vanish at the critical point, acquires a non-vanishing expectation value due to the finite correlation length. However, such value systematically decreases as the correlation length increases, as

$$\left\langle \hat{O}(\xi) \right\rangle \propto \xi^{-x_O/\nu}. \tag{5}$$

The above relation can be used in practice to extract the critical exponent by analyzing the dependence of the expectation value of an operator with respect to the correlation length in what is known as the finite-entanglement scaling approach [14].

Besides standard critical exponents, most of the predictions obtained from Conformal Field Theory [38] (CFT) about the universal properties of a critical system can be accessed through finite-entanglement scaling. Such CFT predictions are usually obtained by using conformal maps, mapping the CFT from the plane to an appropriate geometry, encoding the property one wants to study. For example, one of the most celebrated and important predictions of CFT is that the spectrum of the system in finite geometries is dictated by the critical exponents of the theory, the bulk exponent for systems with periodic boundary conditions (PBC), and the boundary exponents for systems with open boundary conditions (OBC) [39, 40], as obtained by mapping the (upper-half of the) infinite plane to an infinitely long cylinder (strip) with finite circumference (width):

$$\Delta_j^{\text{PBC}}(L) \equiv E_j^{\text{PBC}}(L) - E_0^{\text{PBC}}(L) = \frac{2\pi\nu}{L} x_j, \tag{6}$$

$$\Delta_j^{\text{OBC}}(L) \equiv E_j^{\text{OBC}}(L) - E_0^{\text{OBC}}(L) = \frac{\pi\nu}{L} h_j, \tag{7}$$

where $\nu$ is the speed of sound, and $x_j/h_j$ are the universal scaling dimensions, determined by the operator content of the corresponding CFT and boundary CFT (BCFT), respectively.

On the other hand, universal properties can also be detected through entanglement properties. The scaling of the entanglement entropy of half-infinite segments in a critical spin chain involves the central charge $c$ of the corresponding CFT as [41, 42]

$$S(\xi) = \frac{c}{6} \log(\xi) + \text{const.}, \tag{8}$$

where $\xi$ is the correlation length of the system. In addition to the central charge, the appearance of operator content from a BCFT in the entanglement Hamiltonian $H_{\text{ent}}$ (defined below) was first observed numerically [43] and then derived analytically [44],

$$\rho_A = e^{-2\pi H_{\text{ent}}},$$
$$\lambda_n = \frac{\pi}{W} h_n, \tag{9}$$

where $W \propto \log(\xi)$ is the width of the strip to which the reduced density matrix is mapped after an appropriate conformal mapping, and $h_n$ is the conformal weight corresponding to the $n$-th eigenvalue.[1] Furthermore, the eigenvalues of the entanglement Hamiltonian, $\lambda_{\text{ent}}$, can be analyzed by the finite-size scaling on the strip then if such a conformal mapping exists [44].

In the following, we recite several applications of finite-entanglement scaling for extracting the CFT predictions. Critical exponents were first extracted in Refs. [13, 14]. while the scaling of the entropy was first observed in [14]. The entanglement spectrum has been used

---

[1]Here, the spectrum of the entanglement Hamiltonian is proportional to that of the critical Hamiltonian with open boundary condition as Eq. (9) because there is a conformal mapping to a strip geometry when $\rho_A$ is represented in the path integral.

to identify the operator content in Ref. [22] and subsequently better characterized in terms of the symmetry and operator content in Ref. [31].

In Ref. [22] it has been noted that the low-energy spectrum (extracted from a temporal MPS) and the transfer matrix spectrum extracted from the standard MPS are proportional to each other. This fact, together with the previous observations, see e.g., [16, 22, 45], showing that one could not recover the MPS transfer matrix from known spectra of boundary CFTs is one of the remaining open puzzles about finite entanglement scaling. Similar observations have been made in Ref. [35] where they extract with high precision the speed of sound. There, the MPS transfer matrix spectrum was observed to be proportional to the spectrum of the DMRG effective Hamiltonian, known to describe the low energy excitations [46]. The puzzle is thus manifest—the low energy effective Hamiltonian obtained on the tangent space to the infinite MPS in the finite entanglement regime does not encode the spectrum predicted from CFT in Refs. [39, 40]. We start by reviewing the results we obtain in the finite-entanglement regime for the MPS transfer matrix spectrum.

## 2.1 The MPS transfer matrix spectrum

The transfer matrix (TM) plays an important role in statistical mechanics, in particular in integrable models and their analytical solutions. Beginning with the exact solution of the classical Ising model in 2D by Lars Onsager [47], transfer matrices in translationally invariant models have since also been connected to the Algebraic Bethe Ansatz [48]. In the context of MPS, the transfer matrix plays a crucial role as it quantifies the correlation length in translationally invariant systems in the thermodynamic limit [36]. In those systems, the TM captures the long-range, low-energy behavior of $n$-point correlation functions. Naturally, this raises the question of what the low-energy effective field theory describing the low-energy sector of the MPS transfer matrix spectrum is. In this context, it has been later observed numerically that the spectra of the MPS transfer matrix (3) and the effective low energy Hamiltonian $\hat{H}_{\text{eff}}$ are proportional to each other at continuous phase transition points in the low-energy regime, and we hence consider them interchangeably [22, 35]. This may be intuitively understood from the emergent Lorentz invariance at a critical point, which relates translations in space and time to be the same up to the constant speed. Note that $\hat{H}_{\text{eff}}$ is defined by the iMPS (2) and the matrix product operator of a given Hamiltonian $\hat{H} = \prod_i W_i$ and reads in graphical notation as [46],

$$
\hat{H}_{\text{eff}} = \quad \cdots
\begin{array}{c}
\cdots \; \bar{A} \; \bar{A} \qquad \bar{A} \; \bar{A} \; \cdots \\
\cdots \; W \; W \; W \quad W \; W \; W \; \cdots \\
\cdots \; A \; A \qquad A \; A \; \cdots
\end{array}
\tag{10}
$$

The observation that the MPS transfer matrix has a well-defined, scale-invariant fixed spectrum originally observed in [16, 22, 45] can be rephrased by saying the effective Hamiltonian $\hat{H}_{\text{eff}}$ of the iDMRG algorithm [46] exhibits a scale-invariant ratio of its low-lying energy spectrum akin to that of an RG fixed point. The spectral ratios of the iDMRG MPS are however different from those of the corresponding CFT. Nevertheless, explaining the scale-invariance and the ratio itself has so far been elusive.

Let us start by studying the spectral ratios, numerically obtained by iDMRG, for the critical transverse-field Ising model described by the Hamiltonian

$$
H = - \sum_{n=-\infty}^{\infty} \left( \sigma_n^x \sigma_{n+1}^x + \sigma_n^z \right) .
\tag{11}
$$

As this system is translation invariant, we can describe it through an infinite MPS (iMPS) characterized by a single elementary tensor with finite bond dimension $\chi$ as in Eq. (1). We

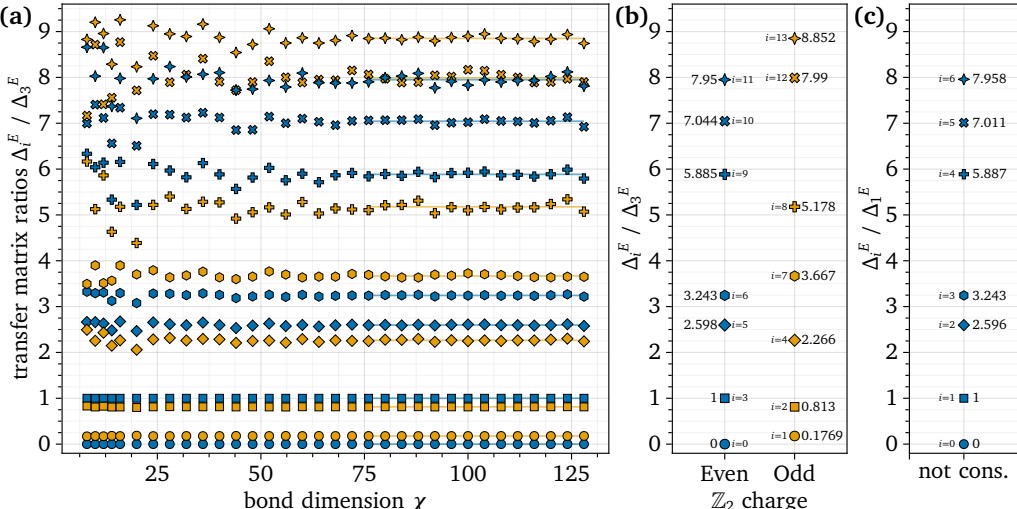

Figure 2: Transfer matrix spectral values $\Delta_i^E$ defined in Eq. (4) of the iDMRG ground state of (11), normalized to the first excited state in the $\mathbb{Z}_2$ even sector. **(a)** Spectral ratios $\Delta_i^E/\Delta_3^E$ are plotted as a function of $\chi$, whereas values are color-coded given the respective $\mathbb{Z}_2$ charge sector. **(b)** The extrapolated spectral ratios $\Delta_i^E/\Delta_3^E$ obtained from fitting a constant to the finite-entanglement scaling of $\Delta_i^E(\chi)/\Delta_3^E(\chi)$ with $\chi \in [76,\ldots,128]$ are shown in their respective $\mathbb{Z}_2$ charge sector. The error bars are smaller than the marker size. **(c)** The transfer matrix spectral ratios $\Delta_i^E/\Delta_1^E$ obtained though iDMRG *without* imposing $\mathbb{Z}_2$ symmetry at $\chi = 96$, and normalized to the first excited state energy $\Delta_1$. We observe the non-conserving case coincides with the $\mathbb{Z}_2$ even sector of the spectrum obtained when explicitly enforcing the symmetry on the elementary tensors.

then employ the VUMPS algorithm for infinite DMRG on it and find the iMPS representation of the ground state given a fixed $\chi$. Subsequently, we inspect the logarithmically transformed eigenvalues of the transfer matrix $\Delta_i^E$, as defined in Eqs. (3) and (4). To clarify symmetry sectors: Note that the Hamiltonian (11) commutes with the total spin parity in the $x$-direction, $P^x = \prod_{n=-\infty}^{\infty} \sigma_n^z$. Therefore, the spectrum of (11) breaks up into even (eigenvalue 0) and odd (eigenvalue 1) $\mathbb{Z}_2$-symmetry sectors. However, one has to enforce this symmetry explicitly in order for the numerical simulation to respect it. This is because even numerically small but not precisely vanishing contributions in floating point arithmetic can lead to a small symmetry-breaking contribution. In case one wants to resolve the given symmetry sectors, one therefore has to ensure these contributions vanish identically as opposed to approximately. This may be achieved with block-sparse tensor arithmetic. In the following, we consider both cases in the infinite chain, explicitly enforcing $\mathbb{Z}_2$ and the case where $\mathbb{Z}_2$ charges are not conserved.

Figure 2(a) shows these $\Delta_i^E$ for each $\chi$ inspected, and the color coding represents the respective $\mathbb{Z}_2$ charge sector. With increasing bond dimensions, the spectral ratios converge to constants. Figure 2(b) and (c) display the extrapolated spectral ratios from these fitted constants in the case of imposing $\mathbb{Z}_2$ symmetry (b) and when not explicitly imposing the symmetry (c), respectively. Note that in the case of the Ising model, the transfer matrix spectrum is indeed real-valued because the Hamiltonian is real and symmetric, which implies all eigenstates are also real-valued. This fact is certainly not true generically, because the TM spectrum is a feature of the ground state, which is generically complex-valued. Naively, one may expect the effective Hamiltonian to be critical as $\chi \to \infty$. Intriguingly, the spectrum observed at large $\chi$ is not the anticipated $c = 1/2$ CFT spectral ratios of $\Delta_n/\Delta_3 = \{0, 1/4, 3/4, 1,\ldots\}$ obtained, for example by considering the spectrum of a critical Ising model defined on a fi-

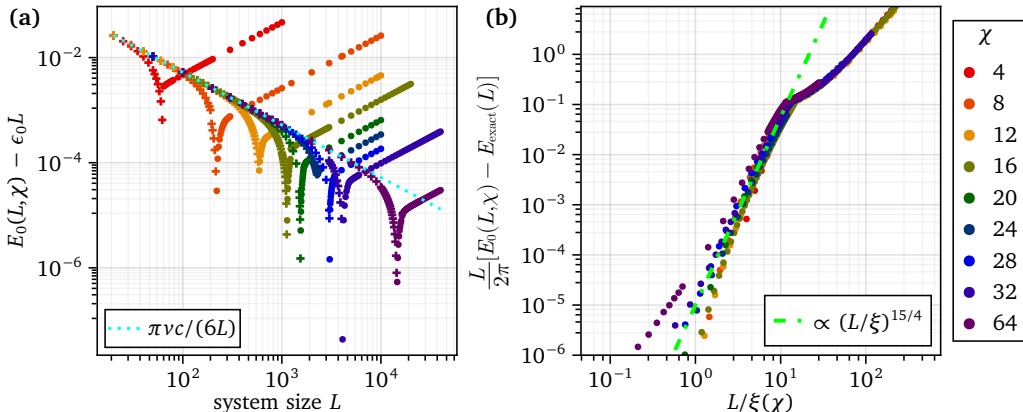

Figure 3: MPS ground state energy $E_0(L, \chi)$ of Eq. (12) shown for several $\chi$ indicated in the legend on the right. **(a)** The ground energy $E_0(L, \chi)$ minus the bulk energy $\epsilon_0 L$ ($\epsilon_0 = -4/\pi$) plotted against $L$ and for various $\chi$. The leading finite-size effect shown in Eq. (13) is plotted with a cyan dotted line. The data points with a negative sign are plotted with "+" markers. **(b)** The re-scaled ground state energy error w.r.t. the exact solution $E_{\text{exact}} = -2/\sin(\pi/(2L))$ is plotted on the $y$-axis. The $x$-axis is the ratio of system size $L$ over the finite correlation length $\xi(\chi) = \xi_0 \chi^\kappa$, where $\kappa(c) = 6/[c(1 + \sqrt{12/c})]$ [15] and $\xi_0 \simeq 0.3$ is set such that the crossover scale denoted with a lime green dotted line matches the data for $L/\xi_0 \chi^\kappa \in [1, 10]$.

nite chain with open boundary conditions described by a BCFT on a strip with free boundary conditions at both edges. The values we obtain are $\Delta_n^E/\Delta_3^E = \{0, 0.1769, 0.8130, 1, \ldots\}$, which also differ from all other known spectra describing the critical Ising model defined on various combinations of finite geometries (rings, finite open chains) and boundary conditions (fixed, twisted, free) described by simple BCFTs. There is also an interesting feature worth mentioning: the symmetry resolved spectrum, obtained by enforcing the $Z_2$ symmetry in the simulation by using explicitly symmetric sparse tensors, is richer than the spectrum obtained by using generic tensors, which do not enforce the symmetry explicitly. The latter spectrum, once rescaled appropriately, only contains the even sector of the symmetry resolved one, as apparent in Figure 2(b) and (c). This fact is clearly reminiscent of a well-known BCFT property, the boundary condition can affect the operator content of the theory [39]. In the following, we will shed some light on this intriguing puzzle, cf. Section 5.2.3.

## 3 The relevant perturbation generating the crossover from the finite-size to the finite-entanglement regime

Keeping in mind what we have described in the previous section, namely that MPS simulations of critical systems with finite bond dimension $\chi$ always entail an effective finite correlation length $\xi(\chi)$, when studying a finite system we have to consider two scales, $\xi(\chi)$ and $L$, the size of the system. If $L$ is smaller than $\xi(\chi)$, the MPS can faithfully simulate the critical system, as the finite-size effects dominate the finite correlation length [49]. However, for $L > \xi(\chi)$, the inherent behavior is controlled by $\xi(\chi)$, which exhibits scaling different from the one expected from the critical point described by a conformal field theory (CFT). This phenomenon is often referred to as a crossover from finite-size to finite-entanglement scaling [14–16].

First, we want to elucidate how this crossover from finite-size to finite-entanglement scaling occurs in the ground state. Therefore, we first study finite system sizes in periodic boundary

conditions before turning towards the thermodynamic limit in the next section. To this end, we study the critical transverse-field Ising (TFI) model defined by the Hamiltonian,

$$\hat{H} = -\sum_{n=1}^{L} \hat{\sigma}_n^x \hat{\sigma}_{(n+1)\%L}^x - \sum_{n=1}^{L} \hat{\sigma}_n^z, \tag{12}$$

where the position $n+1$ is taken modulo $L$ implementing periodic boundary conditions (PBC) using the periodic and uniform MPS (puMPS) algorithm proposed in Ref. [16]. In this case, the algorithm uses non-symmetric tensors, and it thus does not enforce the $\mathbb{Z}_2$ symmetry. Thanks to the translation invariance in the finite system, we can describe it by a puMPS consisting of $L$ copies of a single elementary tensor $A$,

$$\left|\psi_{\text{puMPS}}\right\rangle \equiv \sum_{\{\alpha_i\}} A_{\alpha_1\alpha_2}^{i_1} \cdot A_{\alpha_2\alpha_3}^{i_2} \cdots A_{\alpha_{L-1}\alpha_L}^{i_{L-1}} \cdot A_{\alpha_L\alpha_1}^{i_L} =$$

with the uniform bond dimension $\chi$. For more algorithmic details, we refer to Refs. [16,50]. As the Hamiltonian (12) has an exact analytical solution, the numerical error can be defined as a deviation from that well-known analytical value.

More precisely, we define the numerical error of the puMPS simulation $\delta E_0(L,\chi)$ as,

$$\begin{aligned} \delta E_0(L,\chi) &= E_0(L,\chi) - E_0^{(\text{exact})}(L) \\ &= E_0(L,\chi) + \frac{2}{\sin\left(\frac{\pi}{2L}\right)}, \end{aligned} \tag{13}$$

where the exact solution for the TFI model with PBC [51,52] is employed. In Fig. 3(a), we show the leading finite-size correction to the ground state energy both for different MPS data (rainbow color) and in cyan the analytical leading-order correction encoding the Casimir energy $E_{\text{Casimir}}(L) = \frac{\pi v c}{6L}$ with $v = 2$ the speed of sound[2] and $c = 1/2$ the central charge in the Ising model, respectively [53,54]. We defined $\epsilon_0$ as the bulk energy density in the thermodynamic limit, which is expressed as $\epsilon_0 = \lim_{L\to\infty} -2/(L\sin(\frac{\pi}{2L})) = -4/\pi$. The larger $\chi$, the MPS results follow the predictions for larger $L$. For example, the ground-state energy obtained from MPS with $\chi = 64$ is consistent with the dotted line, up to $L \approx 6\cdot 10^3$. However, as we proceed to larger system sizes, deviations from the theoretical value are observed. These higher order deviations are shown in Fig. 3(b), following a power-law:

$$\frac{L}{2\pi} \delta E_0(L,\chi) \propto \left(\frac{L}{\xi(\chi)}\right)^{15/4}. \tag{14}$$

This power-law increase in energy can be understood as a second-order perturbation from the magnetic operator. To deduce this, let us first elaborate that, in general, the effective Hamiltonian is expressed as

$$\hat{H}(L) = \hat{H}^* + \sum_n \int_0^L \mathrm{d}x \; g_n(L)\hat{\Phi}_n,$$

where $\hat{H}^*$ represents a scale-invariant Hamiltonian described by a CFT, and $\hat{\Phi}_n$ is a perturbative operator. The running coupling constants $g_n(L)$ evolve according to the RG equations:

$$\frac{\mathrm{d}g_n}{\mathrm{d}\ln(L)} = (D - x_n)g_n, \tag{15}$$

---

[2]The speed of quasi-particles, a.k.a. the speed of light. Not to be confused with the cursive Greek letter representing the critical exponent $v$.

with $D$ being the space-time dimension (in this specific case, $D = 1+1$ and $x_n$ the scaling dimension of the corresponding operator $\hat{\Phi}_n$). Solving this equation gives

$$g_n \propto L^{D-x_n}.$$

It is important to emphasize the scale dependence of $g_n$, which is generally referred to as a "running coupling". If the scaling dimension $x_n$ is smaller than $D$, the perturbation $g_n(L)$ is a relevant perturbation in the effective theory, and it hence grows as $L$ increases. This implies that no matter how small the initial perturbation $g_n(L=1)$ is, for example, $g_\sigma(1) = 0.00001$, $g_\sigma(L)$ can reach $\mathcal{O}(1)$ at $L \simeq 400$. The only exception is when the initial coupling is $g_\sigma(1) = 0$, where the system remains critical and scale-invariant. At criticality, the energy gaps take a universal value, expressed in Eq. (7). Thus, the spectral ratios $\Delta_n(L)/\Delta_1(L) = x_j/x_1$ are scale-invariant, up to finite-size corrections. However, the presence of running couplings alters these ratios. In particular, we can use the operator state correspondence of CFT to identify excited states with the CFT operators. If such CFT operators have non-vanishing operator product expansion (OPE) coefficients with the relevant perturbation, they experience an energy correction $\delta E_j(L) = E_j(L) - \frac{2\pi v}{L} x_j$. If such correction occurs at $m$-th order in perturbation theory, $E_j(L)$ is shifted due to $\int g_n \hat{\Phi}_n$ as [39, 55] (for open boundary conditions [56]),

$$\frac{L}{2\pi} \delta E_j(L) \propto g^m(L). \tag{16}$$

In this case, the spectral ratios are no longer scale-invariant for the system perturbed away from the critical point. By analyzing such energy deviation from the universal values, we can infer the scaling dimension of the operator that perturbs the system.

In the case of the Ising CFT, if we do not enforce $\mathbb{Z}_2$ symmetry, the most relevant operator is the magnetic operator $\hat{\sigma}(n) \simeq \sigma_n^x$. We know that due to the spin-flip symmetry, its leading effects in perturbation theory occur in second order. Therefore, the leading correction to the ground state energy from the magnetic operator $\hat{\sigma}$ is,

$$\frac{L}{2\pi} \delta E_0 \propto g_\sigma^2 \propto (L^{2-x_\sigma})^2 = L^{\frac{15}{4}}, \tag{17}$$

given the known scaling dimension $x_\sigma = 1/8$. This result aligns with our numerical observations shown in Fig. 3(b) when replacing the renormalization scale $L$ with $L/\xi(\chi)$ being the relevant length scale in the crossover regime from finite-size to finite-entanglement scaling.

In summary, the previous analysis allows us to confirm that the error induced by using a finite bond dimension MPS has a clear RG interpretation. This error is forced to grow as we fix the bond dimension and increase the system size. This phenomenology is the same that we observe if, rather than finding the ground state of the original Hamiltonian, we study the ground state of the critical Hamiltonian perturbed by a relevant operator. We can thus conclude that the MPS ground state with a finite bond dimension is the ground state of a perturbed Hamiltonian obtained by adding a relevant perturbation to the critical Hamiltonian. Such a Hamiltonian only works at low energies, and thus it is an effective Hamiltonian.

## 4 Puzzling scale invariance: A self-congruent point

Here, we return our focus on the thermodynamic limit by employing iDMRG algorithms on infinite MPS (2) thereby removing the system size's influence on the tensor network. The question is if we can define a fixed point or something akin to a fixed point that correctly describes the finite-entanglement phenomenology in our observed numerics, manifested in the transfer matrix spectrum of an iMPS. As discussed in Eq. (16), $g_r(\chi, L) \propto L^{D-x_r}$ can

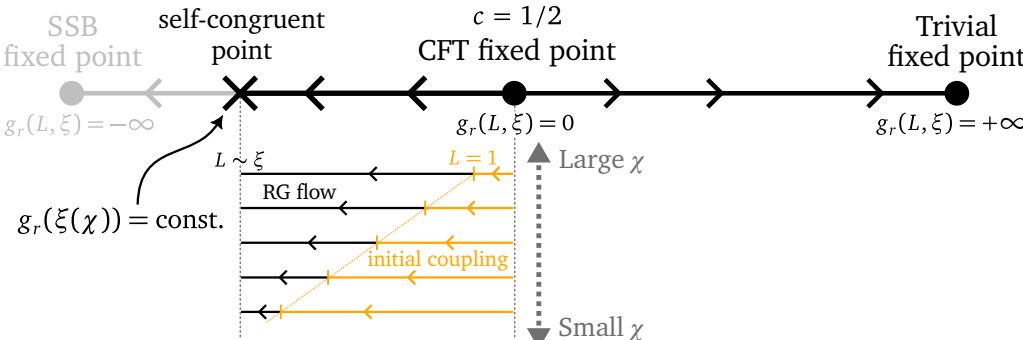

Figure 4: A schematic picture of a self-congruent point in the RG flow of a relevant coupling $g_r$ in the EFT corresponding to the infinite MPS of the TFI model (11). The RG flow is dictated by Eq. (15). Following the flow amounts to increasing the typical length scale $L$ up to the maximum length scale set by the correlation length $\xi(\chi)$ in iDMRG. For small bond dimension $\chi$, the MPS corresponds to a ground state of an EFT with a larger initial (microscopic) coupling than for larger $\chi$, where the corresponding EFT has a smaller initial coupling closer to the critical value. Initialized in the vicinity of the critical point, the RG flow comes to a halt at the *self-congruent point*, at which the coupling becomes a constant as a function of $\chi$, see Eq. (20). In any numerical simulation, the finite entanglement resource (quantified by $\chi$) thus not only inhibits the EFT from being initialized at the critical coupling but also bounds the RG flow up to a finite distance from the initial value of the coupling.

become large with increasing $L$. In the iDMRG case, there is no parameter $L$, but $\xi(\chi)$ acts as the proxy for the system size in the RG flow equations,

$$L \equiv \xi(\chi),$$

as it quantifies the length scale over which degrees of freedom in the infinite tensor network are effectively interacting. This correlation length $\xi(\chi)$ increases algebraically with $\chi$ as [15, 16],

$$\xi(\chi) \propto \chi^{\kappa},$$
$$\kappa(c) = \frac{6}{c(1 + \sqrt{12/c})}.$$

Thus, a larger $\chi$ renormalizes $g_r$ further away from the initial couplings as $\xi(\chi)$ grows, and a larger $\xi$ allows the RG flow to run longer.

In contrast, employing a larger $\chi$ effectively implies that we are simulating a perturbed system with a smaller initial relevant perturbation $g_r(L = 1)$ at the original lattice spacing scale, as we elucidated in Section 3. The effects of this perturbation at different scales are dictated by the corresponding RG equations. As a result, the small initial perturbation drives the system away from the critical point when probed at large scales $L \sim \xi(\chi)$. Consequently, the effects of the perturbation become independent of the scale set through $\chi$, as observed in the TM spectrum of the iDMRG MPS (Fig. 2), since these two opposite effects exactly balance each other out. As a result, we find that there is a point within finite distance from the critical point where the running coupling effectively stops running when studied as a function of the only scale available in translationally invariant MPS, $\chi$,

$$g_r(\xi(\chi), \chi) = \text{const.}$$

This is schematically illustrated in Fig. 4. From this point of view, let us revisit Fig. 3. Noting that the vertical axis corresponds to the amplitude of $g_\sigma^2$, the data collapse in Fig. 3(b) indicates

the following relation:

$$g_\sigma(L,\chi) \propto \left(\frac{L}{\xi(\chi)}\right)^{2-x_\sigma}. \tag{18}$$

This implies a smaller initial running coupling and a constant final perturbation at $L \sim \xi(\chi)$, as shown by:

$$g_\sigma(1,\chi) \propto \left(\frac{1}{\xi(\chi)}\right)^{2-x_\sigma} = \left(\frac{1}{\xi(\chi)}\right)^{15/8}, \tag{19}$$

$$g_\sigma(\xi(\chi),\chi) \propto \text{const.} \tag{20}$$

Equation (20) shares similarities to a genuine fixed point equation, as the RG flow for different $\chi$ leaves the relevant coupling $g_\sigma$ unaltered. This is however only due to the limited correlation length in the infinite MPS, which is acting as a proxy for the system size. The effect of the RG flow in the infinite MPS is thus capped at the scale of the correlation length, thereby introducing a spurious feature akin to an RG fixed point in the sense that the RG flow is unaffected by increasing $\chi$. We hence call this feature a *'self-congruent point'*, as the halting point of the RG flow coincides for different $\chi$, which is in particular within a finite distance to the critical point. The coupling thus effectively stops running and becomes constants w.r.t. the bond dimension $\chi$, and we can thus observe scale invariant ratios. As such, the *self-congruent point* arises from the constraints of finite-entanglement scaling. In particular, it remains located at a finite distance from the CFT fixed point, even when we consider the limit $\chi \to \infty$. This property sharply contrasts with the other known fixed points, the stable (non-critical) RG fixed points. Their distance, mea from the unstable CFT fixed point increases logarithmically as we increase the system size. As a result, it diverges in the thermodynamic limit as we consider the limit $L \to \infty$, so diverges $\log(L)$ towards $\infty$.

Equation (18) is not just a numerical artifact but rather a universal consequence of the approximation based on the MPS with a finite bond dimension near a critical fixed point. Here, the initial coupling constant can be considered proportional to the perturbation $t$ that appears in the lattice model. Near criticality, it is well known that $t$ has a universal relation to the correlation length as:

$$\xi \propto |t|^{-\nu}, \tag{21}$$

where $\nu$ is a universal critical exponent $\nu = 1/(D - x_r)$. This relationship is a direct consequence of the correlation length and the energy gap in field theory, $\Delta \sim 1/\xi$. By substituting $t \sim g_r(L = 1)$ and $\nu = 1/(D - x_r)$ into (21), we can derive Eq. (19). Thus, the apparent scaling invariance occurs because there is no system size for the iDMRG; instead, $\xi(\chi)$ plays the role of the system size in the RG flow equations. This $\xi(\chi)$, in the same way, determines the amplitude of the initial couplings through $1/\xi \sim \Delta \sim g_r(L = 1)$, and this balance renders the spectral ratios independent of $\chi$.

Paraphrasing, the apparent scale invariance is a consequence of the balance of two counteracting factors: the renormalized amplitude of the relevant perturbation $g(\xi, \chi)$ and the correlation length $\xi(\chi)$. The bond dimension $\chi$ is a truncation parameter, and the correlation length increases with $\chi$ as $\xi(\chi) \propto \chi^\kappa$. Larger $\chi$ allows us to simulate a system closer to the original critical lattice model. In terms of the field theory of the effective Hamiltonian, this implies that the initial value for the running coupling constants in the emergent EFT with relevant perturbation $g_r(L = 1, \chi)$ are smaller for larger bond dimensions. This argument is consistent with the argument made in Refs. [15, 16], which we briefly summarize in Appendix A. They argue that the difference of the variational energy per unit length to the exact solution is minimized in iDMRG in the case when the truncation error is proportional to $1/\xi$.

It is important to note that our results imply that the limits $L \to \infty$ and $\chi \to \infty$ do not commute. Specifically, if we first take the limit $\chi \to \infty$ at a fixed $L$, we follow the finite-size scaling approach to reach the CFT fixed point. Conversely, if we first take $L \to \infty$ at

a fixed $\chi$, the result will approach the self-congruent point, even as $\chi \to \infty$. This means that we cannot eliminate the infinitesimal relevant perturbation induced by the finite bond dimension. While this perturbation decreases with increasing $\chi$, the IR scale simultaneously grows with $\chi$, leading to the system always reaching the self-congruent point at the IR scale. This observation, however, is practically irrelevant. As we increase the bond dimension, the RG trajectories of an MPS state with a finite bond dimension become indistinguishable from those of the exact critical ground state over longer segments. These segments are used to extract universal properties from bulk observables. Therefore, the existence of a self-congruent point does not undermine any known result related to finite-entanglement scaling.

## 5 An effective lattice Hamiltonian

In this section, we want to address whether we can explain the symmetry-resolved spectral ratio of Fig. 2 within our framework described above. Since this spectral ratio is beyond perturbative field theory, we instead construct a lattice model that reproduces these spectral ratios for the lowest-lying levels. In particular, we consider a perturbed lattice model

$$\hat{H} = - \sum_{n=1}^{\xi(\chi)-1} \sigma_n^x \sigma_{n+1}^x - \Gamma(\xi(\chi)) \sum_{n=1}^{\xi(\chi)} \sigma_n^z, \tag{22}$$

where we design the transverse field $\Gamma(\xi(\chi)) \to 1$ as $\xi(\chi) \to \infty$ so that this becomes the original transverse field Hamiltonian (12) in the infinite $\chi$ limit, and we move away at any finite $\chi$ by the effect of a thermal perturbation $\varepsilon(n) \simeq \sigma_n^z$, the only one that respects the $\mathbb{Z}_2$ symmetry and thus allow to study a symmetry resolved spectrum. Note that our effective Hamiltonian has open boundary conditions.

The ansatz (22) is the simplest way to perturb the Hamiltonian and is inspired by our numerical observations of the previous section. Despite the fact that the transfer matrix spectrum is not a CFT spectrum, its value 0, 0.1769, 0.8130, and 1.0 displayed in Fig. 2 have the structure expected for a free Fermionic model, since the sum of the first and second excitation energies approximately equals the third,

$$0.1769 + 0.8130 \approx 1.0 \, .$$

Such a phenomenon implies that there are two fermionic modes that can be created acting on the vacuum with its creation operator $c_{k_1}^\dagger |0\rangle$, $c_{k_2}^\dagger |0\rangle$. Each of these states has a specific energy, and the energy of the state one creates by acting with the two operators $c_{k_2}^\dagger c_{k_1}^\dagger |0\rangle$ is the sum of the energy of the single fermion states. This property is also consistent with the $\mathbb{Z}_2$ charge assignment of the low energy spectrum. A single fermion is in the $\mathbb{Z}_2$ odd sector, while two fermions generate $\mathbb{Z}_2$ even states. And we indeed observe that the two lowest states are odd, while the third is even.

Incidentally, Eq. (22) has a rather simple form: Recall that the initial coupling constant $g_r(L=1, \chi)$ is proportional to the perturbation in the lattice model. In the current case, the perturbation respecting $\mathbb{Z}_2$ symmetry has the initial coupling as (see (19))

$$g_r = g_\epsilon(1, \chi) \propto \xi^{-(2-x_\epsilon)} \, ,$$

whereas now $x_\epsilon = 1$. Thus, we shall construct a transverse field Ising model, including a $\mathbb{Z}_2$ symmetry-preserving perturbation of the order of $\sim 1/\xi(\chi)$. To this end, we replace $L \to 2\xi(\chi)$, since $\hat{H}_{\text{eff}}$ (10) is symmetrically constructed with a left and right environment

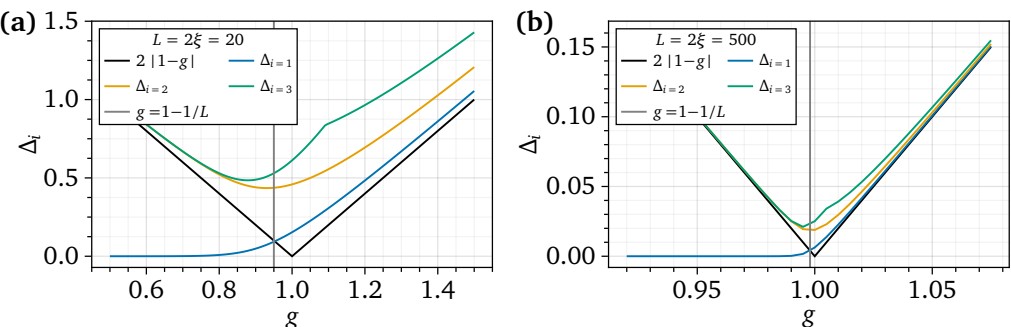

Figure 5: Spectrum of Eq. (23). **(a)** The spectral gaps $\Delta_i$ at $L = 20$ plotted against $g$. The black line marks the single particle gap from Eq. (25), $2|1-g| = 2|\delta|$. **(b)** The same as (a) but for $L = 500$. One observes the gap in the thermodynamic limit is given by $\Delta = 2|1-g|$.

each with a typical length scale of $\xi(\chi)$ induced by the finite bond dimension on either side. We hence obtain an effective Hamiltonian as follows:

$$
\begin{aligned}
\hat{H}(\xi) &= -J \sum_{j=-\xi+1}^{\xi-1} \hat{\sigma}_j^x \hat{\sigma}_{j+1}^x - \Gamma_c \left(1 - \frac{m}{2\xi}\right) \sum_{j=-\xi+1}^{\xi} \hat{\sigma}_j^z \\
&= -J \sum_{j=1}^{L-1} \hat{\sigma}_j^x \hat{\sigma}_{j+1}^x - \Gamma_c \left(1 - \frac{m}{L}\right) \sum_{j=1}^{L} \hat{\sigma}_j^z,
\end{aligned}
\tag{23}
$$

where $L = 2\xi$, $\Gamma_c = J$ is the critical value, and $m$ is a free parameter for now. In the following, we will use the labels $g(\xi) = 1 - \frac{m}{2\xi} = 1 + \delta$ and call the perturbation $\delta = -m/(2\xi) = -m/L$. We will show below that we can fix $m = 1$ and then this Hamiltonian reproduces the lowest-energy spectral ratios of the MPS transfer matrix.

## 5.1 Analytical study of the effective lattice model

The spectrum of Hamiltonian (23) can be determined through a Jordan–Wigner transformation followed by a Bogolyubov transformation [57–59]. The Hamiltonian, after the diagonalization, reads

$$
\hat{H}(L) = E_{\text{gs}} + \sum_k \mathcal{E}_k \hat{c}_k^\dagger \hat{c}_k,
\tag{24}
$$

where $E_{\text{gs}}$ is the ground-state energy, $\hat{c}_i^\dagger$ are fermion creation operators and $\mathcal{E}_i$ are the energies associated with the corresponding fermions. The excited states of this diagonalized Hamiltonian are those obtained by $\hat{c}_{k_1}^\dagger \cdots \hat{c}_{k_l}^\dagger |0\rangle, k_i \geq 1$, with excitation energies $\Delta = \sum_{i=1}^l \mathcal{E}_{k_i}$. The energy of elementary excitations $\mathcal{E}_k$ is given by

$$
\mathcal{E}_{k_i} = 2\sqrt{1 + g^2 - 2g \cos k_i},
\tag{25}
$$

where we define $g = \frac{\Gamma_c}{J}(1 - \frac{m}{L})$ and $k_i$ are the solutions of the equation:

$$
\frac{\sin k(L+1)}{\sin kL} = \frac{1}{g} := \frac{J}{\Gamma_c(1 - \frac{m}{L})} = \frac{1}{1 - \frac{m}{L}}.
\tag{26}
$$

The analysis of the spectrum is much simplified by taking the scaling limit $L \to \infty$ [60]. In fact, the scaling limit corresponds to the universal behavior of the Ising criticality with the thermal perturbation, which is valid beyond the particular choice of the lattice model (23).

First, let us show that the above equation permits an imaginary root of $k$ when $m > 1$. The imaginary root is related to the exponential degeneracy of the ground states in the spontaneous symmetry-breaking phase $g < 1$ [58, 59]. In a finite-size system, the threshold for the appearance of the imaginary root is shifted from $g = 1$ to $m = 1$ in the scaling limit [60], as we will see below. Substituting $k = \frac{ix}{L} + \mathcal{O}\left(\frac{1}{L^2}\right)$ into the quantization condition (26), one obtains,

$$\frac{x \cosh x - m \sinh x}{L} + \mathcal{O}\left(\frac{1}{L^2}\right) = 0 \,, \tag{27}$$

which at leading order requires

$$\tanh x = \frac{x}{m} \,, \tag{28}$$

to be satisfied. Equation (28) permits a nontrivial real solution only when $m > 1$, we denote this solution by $x_1$. So when $m > 1$, the lowest elementary excitation energy is

$$\mathcal{E}_1 = \mathcal{E}\left[\frac{ix_1}{L} + \mathcal{O}\left(\frac{1}{L^2}\right)\right] = \frac{2\sqrt{m^2 - x_1^2}}{L} + \mathcal{O}\left(\frac{1}{L^2}\right) \,. \tag{29}$$

On the other hand, the 2nd lowest elementary excitation is given by the real root of Eq. (26). This time, by substituting $k = \frac{x}{L} + \mathcal{O}\left(\frac{1}{L^2}\right)$ into Eq. (26), one obtains the condition:

$$\frac{x \cos x - m \sin x}{L} + \mathcal{O}\left(\frac{1}{L^2}\right) = 0 \,, \tag{30}$$

which at the leading order amounts to

$$\tan x = \frac{x}{m} \,. \tag{31}$$

In the region $x \in (\frac{\pi}{2}, L\pi)$, the above equation always allows $L - 1$ solutions, in which we denote the smallest one as $x_2$. The solution in the region $(0, \frac{\pi}{2})$ only exists when $m < 1$, which gives the lowest elementary excitation in this case and will be denoted as $x_1$, instead. Overall, the elementary excitation energy corresponds to the real solution of Eq. (26) is always given by,

$$\mathcal{E}\left[\frac{x_i}{L} + \mathcal{O}\left(\frac{1}{L^2}\right)\right] = \frac{2\sqrt{x_i^2 + m^2}}{L} + \mathcal{O}\left(\frac{1}{L^2}\right) \,, \tag{32}$$

which suggests the system approaches a vanishing gap in the $L \to \infty$ limit as expected.

At this point, we want to present the arguments that fix $m$ to actually be $m = 1$. First, we observe from $\mathcal{E}_1(x \to 0) = \Delta_1$ that the single particle gap in the thermodynamic limit is given by $\Delta_1 = 2|1 - g| = 2|\delta|$, which is what one can also observe in Fig. 5, where we plot the numerically exact spectrum of Eq. (23). We observe the first non-degenerate gap approaches $2|1 - g|$ in the thermodynamic limit. Second, we recognize that the correlation length in the spontaneous symmetry-breaking (SSB) phase [61] and the paramagnetic (PM) phase scale equivalently in terms of $|\delta|$, that is $\xi \propto |\delta|^{-1}$, but have different proportionality factors. These proportionality factors between the correlation length $\xi$ and the single particle gap $\Delta_1$ (or conversely the perturbative coupling $\delta$) in the SSB phase and the paramagnetic phase, respectively, can be recovered from the decay of the analytical forms of the correlation functions known from the algebraic Bethe ansatz of the Ising model, cf. Table 1.1 in Ref. [48]. We find the following relations,

$$\xi_{\text{SSB}} \propto (2\Delta_1)^{-1} \,, \qquad \text{for } \delta < 0 \,, \tag{33}$$

$$\xi_{\text{PM}} \propto \Delta_1^{-1} \,, \qquad \text{for } \delta > 0 \,, \tag{34}$$

which can be intuitively understood as follows. In the PM phase, a single quasi-particle excitation creates nontrivial correlations on top of the ground state. In contrast, in the SSB phase, the quasi-particles are domain walls, implying that only two quasi-particles can generate a non-trivial correlation on top of the ground state. The remaining proportionality constant in Eqs. (33) and (34) is given by the speed of sound $v$ of the quasi-particles, which is $v = 2$ in the Ising model [51]. By inserting this proportionality constant $v = 2$, as well as $\Delta_1 = 2|\delta|$ in Eqs. (33) and (34), we obtain,

$$\xi_{\text{SSB}} = \frac{v}{2\Delta_1} = \frac{|\delta|^{-1}}{2}, \qquad \text{for } \delta < 0, \tag{35}$$

$$\xi_{\text{PM}} = \frac{v}{\Delta_1} = |\delta|^{-1}, \qquad \text{for } \delta > 0. \tag{36}$$

In order to show $m = +1$, we only have to first prove that the perturbation induced by the finite bond dimension always settles in the SSB phase as opposed to the PM phase. Then, the claim $m = 1$ follows, because we can conclude with the identification $L = 2\xi$, as argued above Eq. (23), together with,

$$\xi_{\text{SSB}} = \frac{|\delta|^{-1}}{2} = \frac{L}{2|m|}, \tag{37}$$

that $m$ must take values $m = \pm 1$, as well as, $g = 1 \pm 1/L$ in our EFT Hamiltonian (23). The question of which sign $m = \pm 1$ carries is equivalent to the question into which direction away from the critical point does the perturbation settle in the presence of a single[3] relevant perturbation? Below, we give an argument for why the initial marginal perturbation of the transfer matrix and our model points in the direction of the SSB phase, that is why $m = +1$ and $\delta < 0$ in (23). A priori, we know that iMPS have a non-diverging correlation length induced by a finite bond dimension, and we know from RG theory that $\xi \propto |\delta|^{-v}$. Secondly, we know that the iDMRG algorithm, by definition, variationally minimizes the energy density w.r.t. the input Hamiltonian, which is critical. This variational energy density error would be trivially zero at $\delta = 0$ where the ground state is exact and the correlation length diverges. Additionally, the variational energy error is approximately symmetric in $\delta$.[4] Thirdly, there is a fixed correlation length for a fixed bond dimension iDMRG simulation. Therefore, one side of the phase transition is variationally preferred to minimize the energy only if the two sides of the phase transition do not share the same proportionality factor in $\xi \propto |\delta|^{-v}$. This is the case following Eqs. (35) and (36). Given a fixed and finite correlation length $\xi(\chi)$, we find that the amplitudes of the perturbation emerging from such a correlation length on either side of the phase transition relate like,

$$|\delta_{\text{SSB}}| = \frac{|\delta_{\text{PM}}|}{2}.$$

Therefore, iDMRG selects the SSB side since it provides a smaller variational error of the energy density because the error would be trivially minimized at $\delta = 0$, and it is approximately symmetric, cf. reply and attachment at Ref. [62]. Note that such a variational minimization of the energy into the SSB phase does not coincide with the minimization in entanglement entropy, as the PM phase offers in the limit of $\delta \to +\infty$ a trivial fully polarized state with $S_A = 0$, while the SSB phase is double degenerate and one has $S_A = \ln(2)$ in the limit of $\delta \to -\infty$ [62]. The domain wall excitations are generic to SSB phases, and the relation of

---

[3]only a single perturbation with operator $\varepsilon$ due to exact $\mathbb{Z}_2$-symmetry preservation in this Section.

[4]Note that at the phase transition in the Ising model, the variational error has a small bias in favor of the paramagnetic phase, here $\delta > 0$. That is, to leading order, the variational energy is quadratic in $\delta$ with a small bias lowering the variational energy in the PM phase for the same $|\delta|$ in the SSB phase. The dominating effect, however, is the factor of two, which lowers the variational energy significantly in the SSB phase given a fixed $\xi$. See our referee reply and its attachment on SciPost Physics [62].

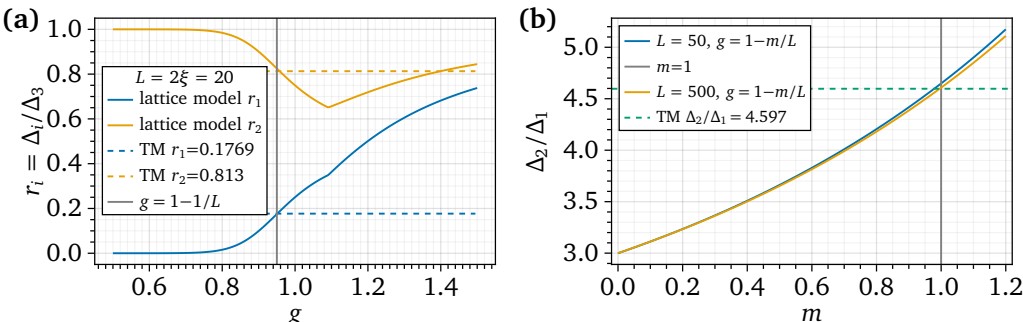

Figure 6: Spectral ratios of Eq. (23). **(a)** The ratios $r_1 = \Delta_1/\Delta_3$ and $r_2 = \Delta_2/\Delta_3$ plotted against $g$ at $\xi = 20$. The MPS transfer matrix ratios, denoted with dotted lines, coincide precisely when $g = 1 - 1/L$ (vertical solid gray line). The phase transition in $\xi = 20$ is located at $g \approx 1.1$. **(b)** Spectral ratio $\Delta_2/\Delta_1$ of Eq. (23) with $\delta = -m/L$ for $L = 50$ and $L = 500$. We observe the well-know ordinary CFT prediction of $\Delta_2/\Delta_1(m = 0) = 3$, and match the TM value of 4.597 at $m \simeq 1$ better in the larger system size.

the single particle gap to the correlation length like in Eqs. (35) and (36) applies to them, too. Therefore, this argument also applies to the critical 3-states Potts model, which we briefly inspect in Appendix C. There, too, we find that iDMRG is selecting for its initial coupling a value in the SSB (ordered) phase, corroborating the above argument.

For general $m$, the spectral ratio $\frac{\Delta_2}{\Delta_1}$ reads

$$
\frac{\Delta_2}{\Delta_1} = \frac{\mathcal{E}_2}{\mathcal{E}_1} = \begin{cases} \sqrt{\dfrac{x_2^2+m^2}{x_1^2+m^2}}, & m < 1, \\[3mm] \sqrt{\dfrac{x_2^2+m^2}{m^2-x_1^2}}, & m \geq 1. \end{cases} \tag{38}
$$

After solving Eq. (31) to obtain $x_2$ in $x \in (\frac{\pi}{2}, \frac{3\pi}{2})$, one may then reproduce the desired ratio at $m = 1$:

$$
\frac{\Delta_2}{\Delta_1} = 4.6033 \approx \frac{0.81304}{0.17687} = 4.59682. \tag{39}
$$

We can furthermore confirm $m = 1$ with our findings of the numerically exact solution of (23) displayed in Fig. 6, where we show in Fig. 6**(a)** the spectral ratios of the transverse field Ising model at $L = 20$. Although the previous discussion was an asymptotic form in $L \to \infty$, the spectral ratio at $L = 20$ has already converged closely to those of the MPS transfer matrix when $m = 1$. These ratios have different values in the thermodynamic limit, depending on $m$. Figure. 6**(b)** displays the spectral ratio of the first two gaps $\Delta_2/\Delta_1$ as a function of $m$. Starting from the ratio predicted by CFT to be $\Delta_2/\Delta_1 = h_{L_{-1}\varepsilon}/h_\varepsilon = 3$, it gradually increases to match that of the MPS transfer matrix at $m \simeq 1$.

Since we have argued in terms of an effective lattice model that has a well-defined continuum limit confined to a finite size of $2\xi$, we follow that the transfer matrix spectrum is universal and not just a peculiar feature of the Ising model. We corroborate this in Appendix B, where we inspect a critical point of the self-dual extended Ising model proposed by Alcaraz [63]. We choose a critical point on the critical line of this model, belonging to the Ising universality class. There, we observe a TM spectrum identical to a very high degree to the one we find in the standard critical Ising model.

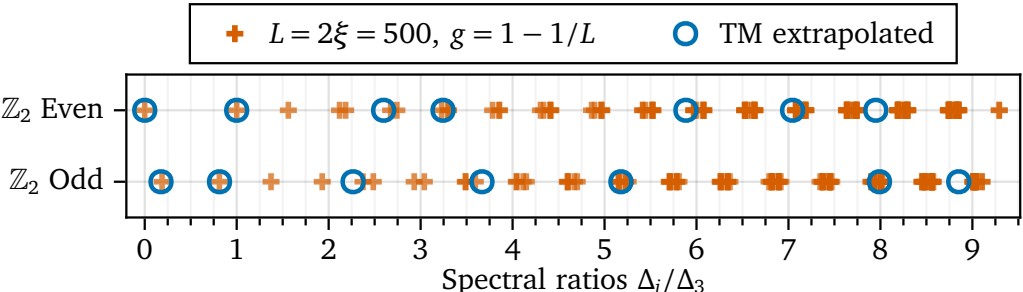

Figure 7: Spectral ratios normalized w.r.t. $\Delta_3$ of the critical TFI model when considering the extrapolated MPS TM values in the thermodynamic limit from Fig. 2 shown as blue circles, and the spectral ratios from the finite-sized Gaussian model (23) with thermal perturbation ($\delta = -1/L$), shown as orange crosses.

## 5.2 Numerical study of the effective lattice model

Here, we numerically compare the features of our effective lattice model (23) to the ones from the MPS transfer matrix, as well as the CFT predictions from the critical point we intend to describe. First, we compare higher spectral ratios of (23) to the ones of the MPS transfer matrix extrapolated in Fig. 2(b). Second, we measure the critical exponents of (23) and compare those to the CFT predictions of the critical Ising model. Third, we inspect the entanglement spectrum and study its convergence towards the expected boundary CFT values.

### 5.2.1 Hamiltonian spectrum

Our findings on the first several hundred spectral values of (23) are displayed in Fig. 7. The spectrum of (23) was computed through the Gaussian fermion ansatz and numerically exactly obtained in a system size of $L = 2\xi = 500$, and is marked as orange crosses in Fig. 7, while the extrapolated values of the TM spectrum form Fig. 2(b) are marked as blue circles. We observe the Gaussian model to fit the MPS transfer matrix spectrum well for the first four energy ratios. All higher energy levels are not matched, as the levels of the Gaussian model are quasi-degenerate levels and much denser packed compared to the levels of the TM spectrum. Therefore, the TM spectrum exhibits Gaussianity in the very lowest-lying energy levels, namely the first four. Contrarily, we conclude that all higher energy states are dominated by non-Gaussian effects, which our Gaussian model cannot capture.

### 5.2.2 Critical exponents

Given that we are introducing a new field theoretical interpretation of the finite-entanglement scaling, we should cross-check that our effective field theory model (23) correctly reproduces the phenomenology observed in this context so far. From our previous discussion, it should be clear that this should be the case, given that the RG running of the system is the RG dictated by the CFT fixed point for all scales smaller than $\xi(\chi)$ where it actually stops running. As a result, all the phenomenology that we have described and that relies on changing $\xi(\chi)$ and observing the response to such a change of scale, should be robust. Below, we shall however confirm our interpretation. We start by extracting the critical exponent in our effective field theory, defined by the "marginally deformed" model in Eq. (23). To this end, we measure the bulk correlation functions $C_{\alpha\alpha}(r) = 4(\langle S_{L/4}^\alpha S_{L/4+r}^\alpha \rangle - \langle S_{L/4}^\alpha \rangle \langle S_{L/4+r}^\alpha \rangle)$ of the ground state of (23) and extract the bulk critical exponents both in the case for $r \ll \xi$ and $r/\xi = $ const. The results for DMRG simulations are displayed in Fig. 8. There, one can observe that for $r \ll \xi$, panel Fig. 8(a), and for $r/\xi = b = 1/2$, panel Fig. 8(b), the critical exponents agree to a high

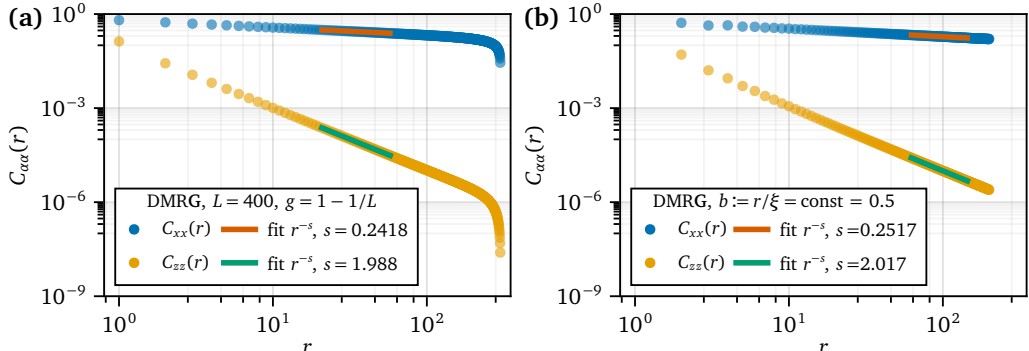

Figure 8: Plots of the connected correlation functions $C_{\alpha\alpha}(r) = 4(\langle S^{\alpha}_{L/4} S^{\alpha}_{L/4+r}\rangle - \langle S^{\alpha}_{L/4}\rangle \langle S^{\alpha}_{L/4+r}\rangle)$ in the ground state of Eq. (23) with $\delta = -1/L$, measuring the decay with distance from the point $L/4$, in order to avoid boundary effects through the OBC. **(a)** $C_{\alpha\alpha}(r)$ with $r \ll \xi$. The ground state was numerically simulated via DMRG at fixed $g = 1 - 1/L$ and $L = 400$. We measure both $C_{zz}(r)$ and $C_{xx}(r)$. **(b)** $C_{\alpha\alpha}(r)$ with fixed $r/\xi = b = 0.5$. The varying ground states therefore follow $g = 1 + \delta(r)$ and are obtained through DMRG simulations, and the position corresponds to the ground state as $\delta(r) = -(r/b)^{-\nu}$ and $L(r) = m|\delta(r)|^{-\nu}$, with $m = 1$ and $\nu = 1$. We simulate $b = 1/2$ and measure $C_{zz}(r)$ and $C_{xx}(r)$. In both cases **(a)** and **(b)**, we find to a high degree the matching bulk CFT critical exponents $x_{\sigma} = s_{xx}/2 \simeq 1/8$ and $x_{\varepsilon} = s_{zz}/2 \simeq 1$.

degree with the well-known bulk CFT values $x_{\sigma} \simeq 1/8$ within 1.3% and $x_{\varepsilon} \simeq 1$ within 0.13%. Figure 8**(a)** furthermore shows a crossover behavior at $r \simeq \xi \simeq L$ with exponentially decaying correlations, implying that the system at scales of the order $\xi$ reveals its gapped nature as expected. The correlation function in Fig. 8**(b)** on the other hand, does not experience such crossover behavior as $r/\xi = b = 1/2 = \text{const.}$ and therefore the size of the system increases with the position $r$ as $L(r) = m|\delta(r)|^{-\nu}$ and $\delta(r) = -(r/b)^{-\nu}$ with $m = 1$ and $\nu = 1$. This suggests that the bulk critical exponents of (23) converge to the standard bulk critical exponents of a $c = 1/2$ CFT with $x_{\varepsilon} = 1$ and $x_{\sigma} = 1/8$ in the thermodynamic limit.

### 5.2.3 Entanglement spectrum

Lastly, we confirm that the entanglement spectrum (ES) of the ground state of Eq. (23) converges towards the expected values from those of the boundary CFT (BCFT) — a fact that was previously known for the iDMRG ground state of Eq. (11) [22, 31]. The expected behavior of the ES therefore contrasts that of the transfer matrix spectrum.

Below, we elaborate the RG behavior of the entanglement spectrum and give reasons for why it converges towards the BCFT spectrum. From the CFT point of view, the underlying reason for the convergence of the entanglement spectrum to the expected values is the conformal mapping that represents the entanglement spectrum as a Hamiltonian,

$$\rho_A = e^{-2\pi H_{\text{ent}}}. \tag{40}$$

The eigenvalues of the entanglement Hamiltonian, $\lambda_{\text{ent}}$, can be analyzed through a conformal mapping, and it turns out to be the BCFT spectrum for gapless cases [44]. For the gapped case, the effective field theory in the coordinates of the entanglement Hamiltonian (Rindler space-time coordinates) can be written as the perturbed action from the (critical) fixed point

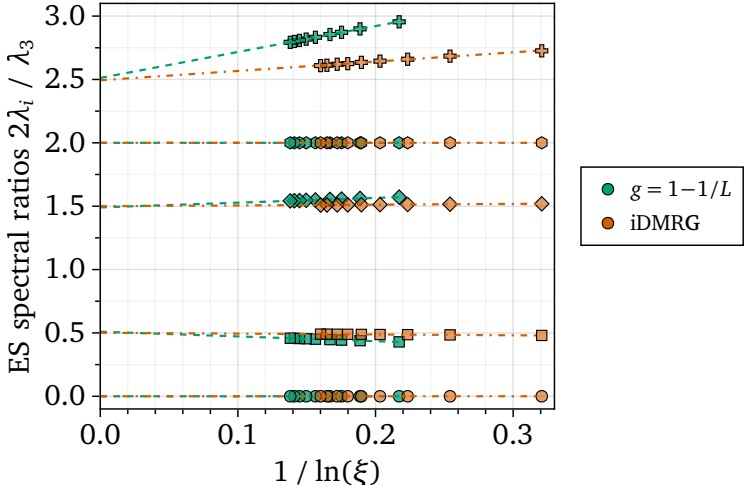

Figure 9: The entanglement spectrum of the ground state of (23) is shown in green markers. The entanglement spectrum of the ground state of (11) obtained by iDMRG ($\chi \in [8, \ldots, 40]$) is displayed as red markers. In the thermodynamic limit, both converge to the boundary CFT prediction of $2\lambda_i/\lambda_3 = \{0, 0.5, 1.5, 2.0, \ldots\}$.

action $S^*$ [64],

$$S = S^* + \delta S, \tag{41}$$

$$\delta S = g(1) \int_0^\infty dx\, e^{(2-\Delta_g)x} \hat{\Phi}_g = \int_0^\infty dx\, e^{(2-\Delta_g)(x-\ln(\xi))} \hat{\Phi}_g, \tag{42}$$

where $\xi \equiv g(1)^{-1/\nu} = g(1)^{-\frac{1}{2-\Delta_g}}$ and the exponential factor arises from the conformal mapping of the primary operator $\hat{\Phi}_g$. Since the perturbation $\hat{\Phi}_g$ is relevant, the RG dimension $(2-\Delta_g)$ is positive. Since the contribution to the partition function is weighted by $e^{-S}$ in the path-integral, the configuration with larger $\delta S$ has a smaller contribution. As a result, the theory is confined to the region

$$0 < x < \ln(\xi).$$

Consequently, the entanglement Hamiltonian is also confined, which has the gapless BCFT spectrum of finite system sizes $\sim \ln(\xi)$ as

$$\lambda_{\text{ent}}^{L \gg \xi} \sim \frac{\pi}{\ln(\xi)}(\Delta_h + n). \tag{43}$$

On the other hand, when $L \ll \xi$, the energy gap induced by the relevant operator is negligible and also becomes the gapless BCFT spectrum as

$$\lambda_{\text{ent}}^{L \ll \xi} \sim \frac{\pi}{\ln(L)}(\Delta_h + n). \tag{44}$$

In the case of our effective model, the entanglement spectrum is an interpolation between two identical free-free boundary BCFT spectra and thus also exhibits the gapless spectrum as Eqs. (43) and (44), where our lattice model corresponds to the intermediate state between two limits characterized by $\xi \simeq L$. Generally speaking, the spectra of Eqs. (43) and (44) are different in the case of iDMRG because an emergent relevant perturbation can, in principle, change the boundary conditions in the effective model towards which the RG flow converges, inducing a boundary RG flow. For instance, if the relevant perturbation is a uniform bulk magnetic field, the emergent physical boundary condition prefers to align with the magnetic

field. This induces the boundary RG flow to free-fixed boundary conditions, as proposed in Ref. [64] and verified in Ref. [31]. Nevertheless, the emergent relevant perturbation induced by finite bond dimension under $\mathbb{Z}_2$ symmetry is $\hat{\varepsilon} \simeq \hat{\sigma}^z$, which flows to free–free boundary conditions. In the context of the transfer matrix spectrum, the boundary conditions on both edges are determined by this emergent physical boundary condition. Thus, the boundary conditions for $\mathbb{Z}_2$ symmetric and non-symmetric cases respectively flow to free-free and fixed-fixed ones. Since the unperturbed BCFT spectrum thereof respectively are $\mathcal{V}_I \oplus \mathcal{V}_\varepsilon$ and $\mathcal{V}_I$, we do not see the spectrum corresponding to $\mathcal{V}_\varepsilon$ in the non-symmetric simulations, while those of $\mathcal{V}_I$ have the same values, as shown in Fig. 2(b) for the even sector and in Fig. 2(c).

Figure 9 shows the entanglement spectrum of our effective Hamiltonian (Eq. (23)) obtained numerically exactly through the Gaussian fermion ansatz (green markers) and through iDMRG (red markers) of the critical Ising model Eq. (11). We observe a clear convergence of both the model and the iDMRG ground state towards the BCFT prediction.

## 6 Conclusion and outlook

In the first part, Section 3, we studied the finite-size energy spectrum of the MPS for the critical Ising model with periodic boundary conditions. Comparing it with the exact solution, we demonstrated that the finite-$\chi$ effects in MPS are caused by $\sigma$, the most relevant perturbation when $\mathbb{Z}_2$ symmetry is not imposed. Our results are consistent with those of Ref. [16], showing that the scaling collapse occurs because the initial coupling constants in the corresponding EFT also scale with the bond dimension.

Moving beyond perturbation theory and considering the thermodynamic limit of MPS, we constructed, in Section 5, an effective lattice model that reproduces the properties observed in iDMRG. This model describes the critical Ising model with a scale-dependent "relevant" perturbation, being marginal in combination with its scale-dependent coupling, where the scale in the infinite MPS is set by the correlation length $\xi(\chi)$. Our effective model accounts for the scale invariance of the iDMRG transfer matrix spectrum, which results from the scale dependence of the perturbation's initial strength. As the bond dimension increases, the scale increases while the perturbation's strength decreases, leading to a freeze-out of the RG flow, effectively halting it because the "relevant" operator was actually marginal in combination with its scale-dependent coupling, cf. Section 3. This behavior results in what we refer to as a *self-congruent point*. The model also explains why the iDMRG transfer matrix spectrum does not match the CFT spectrum, even in the $\chi \to \infty$ limit. While our lattice model reproduces the first non-trivial spectral ratios of the MPS transfer matrix as observed in iDMRG, higher levels deviate, potentially due to non-universal effects. Additionally, our effective model successfully reproduces well-known results from finite-entanglement scaling, including the critical exponent and the entanglement spectrum, as discussed in Section 5.2 [31,64].

Furthermore, we emphasize that in both the critical Ising and 3-state Potts models, the iDMRG MPS encodes the ground state of the system perturbed toward the symmetry-broken (SSB) phase. Due to finite correlation length constraints, this behavior opposes the common expectation that the bipartite entanglement entropy would be minimized at a fixed perturbation strength, as discussed in Section 5.

Our findings highlight the need for further exploration into the finite-entanglement and finite-correlation length scaling of critical systems. Future work should aim to systematically verify these results at other critical points with different symmetries. Additionally, the question of which models and mechanisms explain the higher-energy part of the transfer matrix spectrum remains open.

## Acknowledgments

We are grateful to Rui-Zhen Huang, and Robert Ott for the fruitful discussions. We thank both referees and especially Referee 2 for their constructive, clarifying, and helpful critique. The Gaussian fermionic model was numerically solved with the software package F_utilities [65]. The MPS simulations were in part done using MPSKit [66], as well as using ITensor [67]. This work was initiated at the Workshop "Entanglement Scaling and Criticality with Tensor Networks" held at the Bernoulli Center for Fundamental Studies, EPFL in November–December 2022.

**Funding information**    JTS and LT acknowledge support from the Grant TED2021-130552B-C22 funded by MCIN/AEI/10.13039/501100011033 and by the "European Union NextGenerationEU / PRTR". LT acknowledges support from the Proyecto Sinérgico CAM Y2020/TCS-6545 NanoQuCo-CM, the CSIC Research Platform on Quantum Technologies PTI-001, and from Grant PID2021-127968NB-I00 funded by MCIN/AEI/10.13039/501100011033. AU acknowledges support from the MERIT-WINGS Program at the University of Tokyo, the JSPS fellowship (DC1), BOF-GOA (grant No. BOF23/GOA/021) and Watanabe Foundation. YL is supported by the Global Science Graduate Course (GSGC) program of the University of Tokyo. MO is partially supported by Japan Society for the Promotion of Science (JSPS) KAKENHI Grants JP24H00946 and JP23K25791.

## A    Review on the theory of finite-entanglement scaling

In this section, we review the arguments deriving $\xi \simeq \chi^{\kappa}$ originally proposed by Pollmann et al. [15] and later simplified by Pirvu et al. [16]. The fundamental analytical ingredient for extracting a prediction on the value of the exponent $\kappa$ that dictates the relation of the correlation length and the bond dimension of the matrix product state in the finite entanglement hypothesis is derived by Calabrese and Lefevre in Ref. [68]. Using many analytical assumptions on the analytical structure of the entanglement spectrum in the complex replica plane that are still unverified [69] they derive that the Schmidt values $\lambda_j$ of the critical chain follow a universal decay, characterized by the central charge $c$ and the correlation length $\xi$,

$$n(\lambda) = I_0 \left( 2\sqrt{-b^2 - 2b \ln(\lambda)} \right), \tag{A.1}$$

with $I_0$ the 0-th modified Bessel function, and with,

$$S \propto \frac{c}{6} \ln(\xi), \tag{A.2}$$

$$b := S/2, \tag{A.3}$$

$$b \propto \frac{c}{12} \ln(\xi). \tag{A.4}$$

Firstly, we consider the leading order correction for finite temperature $T$ to the energy density $e$ per unit length at a critical point with conformal invariance to be [53,54],

$$e_0 = e_\infty + \frac{\pi c}{6v} T^2, \tag{A.5}$$

where $e_\infty$ is the energy density of the critical state, and $v$ the speed of the low-energy quasiparticles. In one dimension, the finite, non-vanishing temperature is directly inversely proportional to the correlation length $\xi$ of the system, $\xi \propto 1/T$, which implies that a state with

finite correlation length close to the critical point has an energy density $e$,

$$e_0 = e_\infty + \frac{A}{\xi^2}, \tag{A.6}$$

where $A$ contains non-universal details of the model. We shall denote with $|\psi_0\rangle$ the ground state of the perturbed CFT with finite correlation length and energy density as in Eq. (A.6), and consider its Schmidt decomposition,

$$|\psi_0\rangle = \sum_{n=1}^{\infty} \lambda_n \left|\psi_n^L\right\rangle \left|\psi_n^R\right\rangle, \tag{A.7}$$

where $\left|\psi_n^{L,R}\right\rangle$ are states defined on the left and right semi-infinite half of the system given some arbitrary center point. Normalization of the state obviously requires $\sum_{n=1}^{\infty} \lambda_n^2 = 1$. Note the sum goes all the way up to infinity even in a gapped system, as we are dealing with a field theory and thus infinite degrees of freedom. A finite-$\chi$ MPS now corresponds to a state $|\psi(\chi)\rangle$ with Schmidt decomposition of, crucially, a finite number of Schmidt values (bond dimension),

$$|\psi(\chi)\rangle = \frac{\sum_{n=1}^{\chi} \lambda_n \left|\psi_n^L\right\rangle \left|\psi_n^R\right\rangle}{\sqrt{\sum_{n=1}^{\chi} \lambda_n^2}}, \tag{A.8}$$

while $\lambda_n$ here still refer to the original CFT Schmidt values as in Eqs. (A.1) and (A.7). This definition naturally offers the concept of the residual probability $P_r(\chi)$, defined as,

$$P_r(\chi, b) = \sum_{n=\chi+1}^{\infty} \lambda_n^2 \underset{\chi \to \infty}{=} \int_\chi^\infty \lambda(b, n)^2 \, dn. \tag{A.9}$$

One can show that in the large $\chi$ limit, it takes the form,

$$P_r(\chi, b) = \frac{2b\chi}{\ln\chi - 2b} e^{-b - (\ln(\chi))^2/4b}. \tag{A.10}$$

Furthermore, we can compute the energy density $e(\chi)$ of the approximate state $|\psi(\chi)\rangle$ to the perturbed state $|\psi_0\rangle$ given their fidelity per unit length $f = |\langle\psi_0|\psi(\chi)\rangle|^{2/L}$ and the exact ground state energy density of $|\psi_0\rangle$ denoted as $e_0$,

$$e(\chi) = e_0 f + e_{\text{ex}}(1 - f) = e_0 + (e_{\text{ex}} - e_0)(1 - f). \tag{A.11}$$

Note, that the energy density difference scales as the gap, $e_{\text{ex}} - e_0 \sim \Delta \sim 1/\xi$. The fidelity per unit length is easily understood from the comparison of the Schmidt decomposition to be,

$$f = \sum_{n=1}^{\chi} \lambda_n^2, \tag{A.12}$$

$$1 - f = P_r(\chi), \tag{A.13}$$

as this truncation occurs in an MPS on every unit length once. Consequently, we find,

$$e(\chi) = e_0 + \frac{\beta}{\xi(\chi)} P_r(\chi) = e_\infty + \frac{A}{\xi(\chi)^2} + \frac{\beta}{\xi(\chi)} P_r(\chi), \tag{A.14}$$

where $\beta$ contains the non-universal details of the model related to the gap. Note that there is an optimal value of $\xi$ minimizing the local energy difference $e(\chi) - e_\infty$ for any given $\chi$. In case of large $\xi$, $A/\xi^2$ is small compared to $\beta P_r(\chi)/\xi$; Large $\xi$ implies a large $b$ and a very slow

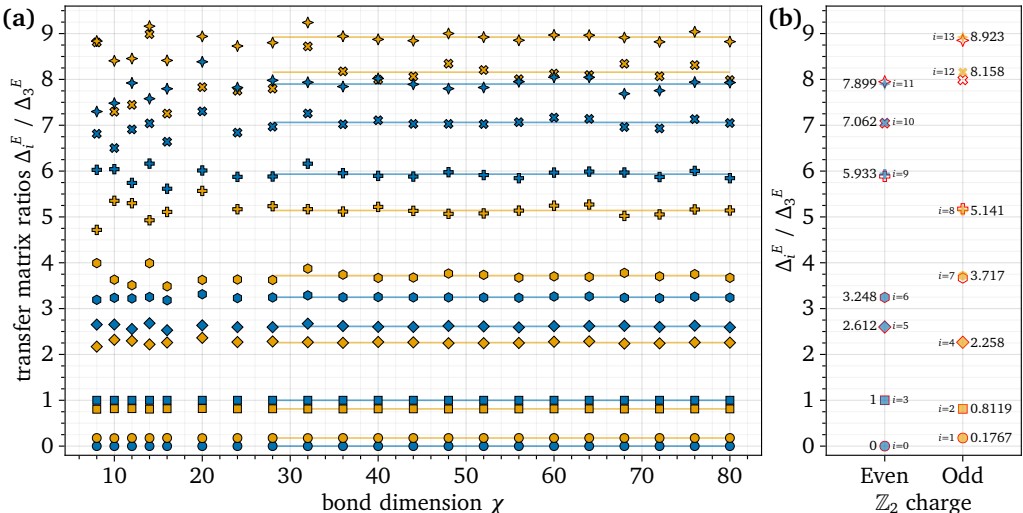

Figure 10: The transfer-matrix spectrum of the iMPS ground state of Hamiltonian (B.1) at $g = 1$ and $p = 0.3$, normalized to the first excited state in the even sector $\Delta_3^E$. **(a)** The TM spectrum is plotted against the bond dimension $\chi$ and the color-coding represents the respective $\mathbb{Z}_2$ charge sector. The constant fit is indicated by the respective line and fitted over $\chi \in [28, 80]$. **(b)** The extrapolated values are plotted against their respective $\mathbb{Z}_2$ charge sector. The extrapolated values of the TM spectrum in Fig. 2, i.e., the standard critical Ising model (11), are overlaid with a transparently filled marker and a red stroke color. One can observe only for higher energy levels a visible discrepancy, while the discrepancy for the ratios $i = 1$ and $i = 2$ is 0.08% and 0.14%, respectively.

decay of $\lambda_n$ in the Schmidt spectrum. The residual probability $P_r(\chi)$ after discarding a finite number of $\lambda_n$ is hence very large. Contrarily, when $\xi$ is small, the relative weight of these two terms flips as opposed to the above scenario. There is hence an optimal $\xi(\chi)$ minimizing the energy difference. The minimum is found to be [15],

$$\xi = \chi^\kappa, \tag{A.15}$$

$$\kappa = \frac{6}{c\left(\sqrt{\frac{12}{c}} + 1\right)}. \tag{A.16}$$

In the $\chi \to \infty$ limit, the minimum of Eq. (A.14) exists only when,

$$P_r(\chi) \sim \frac{1}{\xi(\chi)}, \tag{A.17}$$

which leads to the scaling behavior of the energy density error,

$$e(\chi) - e_\infty \propto \frac{1}{\xi^2}. \tag{A.18}$$

## B   Self-dual extended Ising model

In this appendix, we inspect the self-dual extended Ising (SDEI) model introduced by Alcaraz in Ref. [63], defined by the Hamiltonian,

$$H_{\text{SDEI}}(g, p) = -\sum_{n=-\infty}^{\infty} \sigma_n^x \sigma_{n+1}^x + g\sigma_n^z + p\left(\sigma_n^x \sigma_{n+2}^x + g\sigma_n^z \sigma_{n+1}^z\right), \tag{B.1}$$

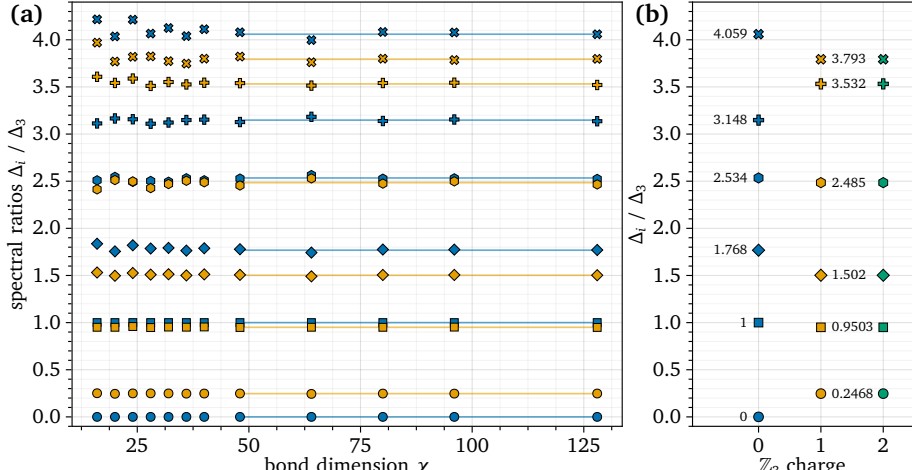

Figure 11: Transfer matrix spectrum of the ground state of the critical three-states Potts model (C.1). **(a)** The spectral ratios $\Delta_i/\Delta_3$ plotted against the bond dimension, and normalized to the first excited state in the sector with $\mathbb{Z}_3$ charge 0, $\Delta_3$. **(b)** The fitted extrapolation of the values in panel (a) over $\chi \in [48,\ldots,128]$, resolved against their respective $\mathbb{Z}_3$ charge. Charge sector 2 and 3 coincide.

which is at $g = 1$ and for any $p < p_c$ critical and the fixed point is in the Ising universality class [63]. The above Hamiltonian is self-dual under the duality mapping,

$$\rho_{2i-1}^{(o)} = \sigma_i^z \sigma_{i+1}^z, \qquad \rho_{2i}^{(e)} = \sigma_i^x, \tag{B.2}$$

where $i = 1, 2, \ldots$, and the new operators obey canonical commutation and anti-commutation relations of spins [63], such that the Hamiltonian (B.1) obeys [63],

$$H_{\text{SDEI}}(g, p) = g\, H_{\text{SDEI}}\left(\frac{1}{g}, p\right), \tag{B.3}$$

up to a boundary term negligible in the thermodynamic limit. We find for $p = 0.3$, the model flows towards the critical Ising model fixed point and exhibits the same transfer matrix spectrum as in Fig. 2, which we show in Fig. 10. The two extrapolated spectra agree to high degree with each other, cf. Fig. 10(b). We conclude that the TM spectrum is indeed universal and a feature of the effective field theory at the critical fixed point, rather than a peculiar feature of the Ising model (11).

## C  Critical three-state Potts model

In this appendix, we inspect the scaling of the MPS transfer matrix spectrum in the critical three-state Potts model defined by the Hamiltonian,

$$H = -\sum_{j=-\infty}^{\infty} \left[ \tilde{Z}_j^\dagger \tilde{Z}_{j+1} + \tilde{Z}_j \tilde{Z}_{j+1}^\dagger + g\left(\tilde{X}_j + \tilde{X}_j^\dagger\right) \right], \tag{C.1}$$

where $Z$ and $X$ are $3 \times 3$ clock matrices, $X^2 + Z^2 = \mathbb{1}$, and

$$\tilde{X} = \begin{pmatrix} 0 & 1 & 0 \\ 0 & 0 & 1 \\ 1 & 0 & 0 \end{pmatrix}, \qquad \tilde{Z} = \begin{pmatrix} 1 & 0 & 0 \\ 0 & \omega & 0 \\ 0 & 0 & \omega^2 \end{pmatrix}, \tag{C.2}$$

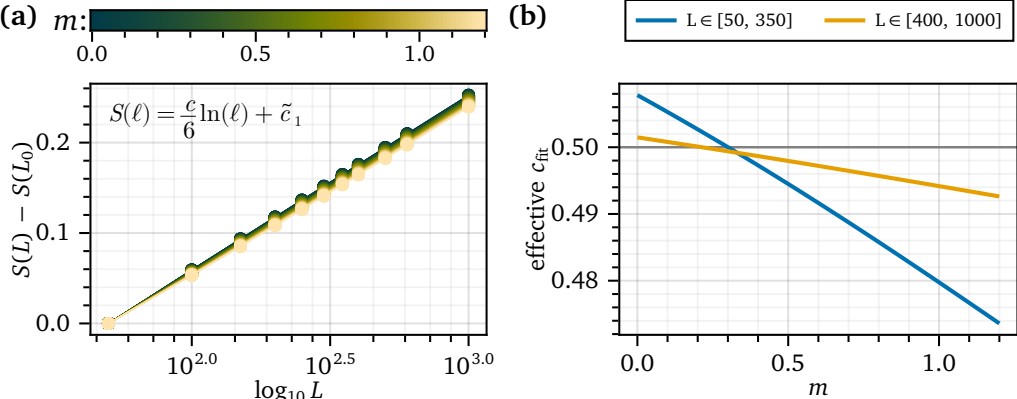

Figure 12: **(a)** Half-chain von Neumann entanglement entropy $S(L)$ in the ground state of Eq. (23) for $\Gamma_c/J = 1$ and system size $L = \xi$ **(b)** The effective central charge $c_{\text{eff}}$ as extracted from the fit of the finite-size scaling of $S(L) - S(L_0)$. One can observe that for larger system sizes $L \to \infty$, the model of Eq. (23) tends towards the Ising CFT with $c = 1/2$ for all values of the tested $m$.

where $\omega = -1/2 + i\sqrt{3}/2$ such that they fulfill the on-site commutation relation $Z_j X_j = \omega X_j Z_j$. Hamiltonian (C.1) is critical at $g = 1$. The model exhibits an ordered phase with spontaneous symmetry breaking when $0 < g < 1$, and a disordered phase when $g > 1$ We show in Fig. 11(a) the transfer matrix spectrum plotted against the bond dimension. Here, too, we observe the self-congruence which leads the TM spectral values to become independent of $\chi$. We extract these constants through fits and resolve their values against their respective $\mathbb{Z}_3$ charge in Fig. 11(b). In the thermodynamic limit and at criticality, Hamiltonian (C.1) is described by a conformal field theory with central charge $c = 4/5$. Its spectral values are given by the conformal weights and read as $\Delta_i/\Delta_3 = [0, 1/3, 5/6, 1, \ldots]$. We observe in Fig. 11(b) that the spectral first gap in the $\mathbb{Z}_3$ charge 1 and 2 sector is shifted below $1/3 = 0.\overline{3}$, while the second gap is shifted above standard CFT value $5/6 = 0.8\overline{3}$, respectively. This suggests that the relevant perturbation in the EFT describing the iDMRG MPS of the critical 3-states Potts model is also taking values in the SSB phase since the levels are shifted towards a 3-fold degeneracy.

## D Finite-entanglement scaling of marginally deformed Gaussian fermions

In this appendix, we consider the lattice model of Eq. (23) and study its Gaussian ground state and its entanglement scaling for finite system sizes $L = \xi$. We aim at characterizing the CFT towards which the Hamiltonian of Eq. (23) tends in the thermodynamic and continuum limit. To this end, we employ the celebrated Calabrese–Cardy formula for the half-chain ground-state entanglement entropy of a CFT in open boundary conditions [70],

$$S(\rho_A) = \frac{c}{6}\ln(\ell) + \tilde{c}_1, \qquad \text{with } \|A\| = \ell, \tag{D.1}$$

in order to fit the effective central charge of the model in a finite system size. Note $\tilde{c}_1$ is a non-universal constant. As discussed in Section 5, the spectrum of (23) is gapless regardless of the value of $m$. We consequently characterize the central charge $c$ by means of finite-size scaling of the entanglement entropy in the ground state $S(L)$ utilizing the free fermion representation of Hamiltonian (23) and the numerical toolkit of Ref. [65]. Our results are shown in Fig. 12, where one can see in panel Fig. 12**(a)** the finite-size scaling of $S(L) - S(L_0)$ against the system

size $L$. The straight line in a lin-log plot corroborates the critical nature of the Hamiltonian and allows us to fit the central charge $c_{\text{eff}}$, which we display in panel Fig. 12**(b)**. One can clearly make out that, for larger system sizes, $c_{\text{eff}}$ tends towards the constant $c_{\text{eff}} = 1/2$. We conclude that in the thermodynamic limit, (23) approaches the Ising CFT with $c = 1/2$, regardless of the value of $m$.

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
