# Peer review of "Self-congruent point in critical matrix product states: An effective field theory for finite-entanglement scaling"

_SciPost Physics, doi:SciPost Phys. 18, 142 (2025)_

## Round 1 · Referee Report · Anonymous (Referee 1) · 2025-1-8

Strengths

The new insight, self-congruent point is introduced to the critical matrix product.

Weaknesses

It is difficult to find out major weakness in this article.

Report

Low energy spectrum of spacial transfer matrix created from variational matrix product (MPS) is theoretically investigated at the criticality of one-dimensional quantum systems. In the case of the transverse-field Ising model (Eq.(11)), it is explicitly shown that the spectrum is quantitatively explained by that of finite size system under the presence of perturbation on the external field (Eq.(23)). The phenomenon is consistently explained by the concept of "self-congruent point", which is realized by the fine balance of the energy resolution and the effective size of the system, regardless of the number of auxiliary state contained in the MPS. These contents are presented step by step with sufficient explanations. As a report introducing new concept on the critical MPS, I recommend the publication of this article (even) as it is. Authors can optionally consider the following points.

(a) The vertical axis of Fig.3 (a) is E_0(L,x)-e_0L, and it takes a couple of time to identify the meaning of e_0, since different form is used in the main text.

(b) The abbreviation "SSB" is used many times. It is relatively odd that the corresponding term is symmetry-broken phase.

Recommendation

Publish (surpasses expectations and criteria for this Journal; among top 10%)

  • validity: high
  • significance: high
  • originality: high
  • clarity: top
  • formatting: excellent
  • grammar: perfect

Author:  Jan Thorben Schneider  on 2025-02-25  [id 5244]

(in reply to Report 1 on 2025-01-08)
Category:
answer to question
correction

We thank the referee for their effort and taking their time to carefully read and critique our manuscript. As their critique is ultimately improving the quality of the manuscript, we are grateful for their comments. Regarding the points raised, we have the following replies:

(a) The vertical axis of Fig.3 (a) is E_0(L,x)-e_0L, and it takes a couple of time to identify the meaning of e_0, since different form is used in the main text.

We have included a more prominent definition of $\epsilon_0$ in the caption, and in the main text which discusses Figure 3 below Eq. (13).

(b) The abbreviation "SSB" is used many times. It is relatively odd that the corresponding term is symmetry-broken phase.

We have updated the first time we used the abbreviation "SSB" to refer to "spontaneous symmetry-broken" phase, on page 17.

The updates are uploaded to the arXiv and are rendered visible to the public within the next publication cycle from the time of this post. Please find attached a document with the complete list of changes, which are also in response to the second referee's critique.

Kind regards The authors.

Attachment:

list-of-changes.pdf

---

## Round 1 · Referee Report · Anonymous (Referee 2) · 2025-1-22

Report

Summary of what I understood from the paper:

The authors address an open problem related to MPS representations of critical ground states. Due to their enforced area law, finite bond dimensional MPS cannot represent such states without inducing an effective correlation length, that increases as does the bond dimension (b. d.). The core concept of finite entanglement scaling theory is the universality of the behavior, in the large b. d. regime, of certain quantities as functions of the correlation length. This is governed by the data of the CFT describing the critical point, similarly to the universality displayed at phase transitions described by perturbations to the CFT. An example of such a quantity would be the spectral ratios of the entanglement Hamiltonian of a half-chain, which converge to the spectrum of scaling dimensions of a BCFT. However, the spectral ratios of the transfer matrix of the MPS converge to values that could not be identified as the spectrum of any BCFT. The authors propose an explanation for this relying on the fact that (i) the TM spectrum is a quantity defined at the scale of the correlation length, and (ii) the UV value of the perturbation due to the finite b.d. decreases when the b. d. increases. In the RG picture, this implies that the effective theory for the TM spectrum is independent of the b.d., being the result of a longer RG flow but for a smaller initial coupling, two effects which cancel out. To provide evidence for this proposal, the authors introduce a model consisting of a family of Ising-like Hamiltonians (all easily diagonalizable by free fermions) with system size-dependent couplings, which converge to the critical Ising model (i.e. the size of the UV perturbation decreases) as the system size (a proxy for the correlation length) increases. These Hamiltonians, meant to represent the effective iDMRG Hamiltonians for different values of the correlation length, show the same spectral ratios as the TM of the Ising model, while their ground states still present the expected CFT governed behavior in their 2-point correlators, entanglement entropy and entanglement spectra.

Report:

I find the ideas in the paper to be interesting and of value and I think they should be published. However, I feel that parts of the paper are hard to follow (or it could be my lack of specific background in the area), and thus I would be more comfortable accepting the manuscript after the authors clarify a bit more some of their arguments and motivations, as per what follows.

To begin with, I missed the motivation about why we should care about the transfer matrix spectral ratios. They are only said to be proportional to the effective DMRG Hamiltonian (although it is not very clear if this is a mathematical result or an empirical observation) and thus related to the low-energy excitations (of the original Hamiltonian?). I guess this correspondence is also the motivation to consider the TM spectrum an observable at the correlation length scale, or is there another argument for that?

The first part of Section 3 is quite nice. I am somewhat confused, nevertheless, by the notion of effective Hamiltonian after Eq. (14). Is it a Hamiltonian whose ground state is the finite b. d. MPS, as opposed to the actual ground state of (12).?What is the relationship between \hat{H} and \hat{H}^*? I also wonder why the choice was made to use the nonsymmetric tensors in this section (leading to a perturbation by sigma) when symmetric ones are used in the following section (leading to a perturbation by epsilon), although I actually don't mind it because that way we get to see both cases.

Right at the beginning of Section 5, what does it mean that "this spectral ratio is beyond perturbative field theory"? In this Section, I understand the second argument given to set m=1 (namely that it matches the numerics that we want the model to reproduce), but the first one (33)-(35) is a bit cryptic, I do not know what assumptions are being made in order to be able to use those formulas. Similarly for the argument before 5.2. that says that the same correlation length "costs" twice the energy when produced in the direction of the paramagnet.

Also, was it checked somewhere that the correlation length of the ground state of H(L) goes as L (i.e. that nu is 1 under Figure 8) or is it an assumption?

Finally, in general I find the notation and the reasoning in Appendix A a bit confusing. We assume that there is a perturbation that generates a correlation length, and then we incur in an error from truncating to finite bond dimension, but if the perturbation that induces the correlation length is the finite bond dimension itself, then there should not be an error due to truncation since psi_0 is already the ground state of the perturbed system, with the finite bond dimension in place, should it?

Some more minor comments are included in the "Requested changes" section

Requested changes

  1. On page 3, the sentence "This has been the subject of several studies" is not followed by a reference.
  2. Beginning of page 5, is "ratio of the gap ratio" the correct expression?
  3. For the unfamiliar reader, it could help to refresh what the scaling hypothesis is. E.g. page 6, "by virtue of the scaling hypothesis, all universal quantities only depend on the correlation length". Is it all there is to the scaling hypothesis, or is it a consequence of a more fundamental assumption?
  4. In Fig. 2, the non-symmetry enforcing case is called symmetry broken instead. Is this intentional? Does the Hamiltonian actually have a degeneracy?
  5. The text after Eq (13) is a bit misleading: this equation represents the difference between the puMPS and the finite size energy, while the plot shows the difference between the puMPS value and the infinite size energy.
  6. On page 13, "transitionally" should read "translationally".
  7. Typo in the caption of Fig 6, 1-1/xi instead of 1-1/L.
  8. On page 21, "the theory is confined to the region", I think I know what is meant: in the spatial region where dS blows up, fluctuations of any field are suppressed and thus effectively log xi is a boundary, i.e. an interface with the vacuum. I would verbalize it a bit more, though.
  9. On page 22,"effectively halting it as the relevant operator was marginal Section 3" is missing an "if" and brackets? 10 . Right before A.11, should ket(psi) be ket(psi_0) everywhere? Also e_0 and e seem to be the same from (A.6),
  10. In appendix B, the model is only KW invariant at g=1 (also the transformation would more accurately be represented as sigma_Z -> sigma_X sigma_X and vice versa). What is the value of p_c?

Recommendation

Ask for minor revision

  • validity: -
  • significance: -
  • originality: -
  • clarity: ok
  • formatting: perfect
  • grammar: excellent

Author:  Jan Thorben Schneider  on 2025-02-25  [id 5245]

(in reply to Report 2 on 2025-01-22)
Category:
answer to question
correction
pointer to related literature

We thank the referee for their effort in taking their time to carefully read and critique our manuscript. Ultimately, their remarks improve the manuscript for which we are grateful. Below, we want to answer all questions the referee raised and hope to do so in a satisfactory way. Furthermore, we have updated the arXiv preprint to v2 implementing several changes to the manuscript which address the requested changes. The replacement is scheduled to be announced at Wed, 26 Feb 2025 01:00:00 GMT.

Please also find the complete list of changes in the PDF file attached.

To begin with, I missed the motivation about why we should care about the transfer matrix spectral ratios. They are only said to be proportional to the effective DMRG Hamiltonian (although it is not very clear if this is a mathematical result or an empirical observation) and thus related to the low-energy excitations (of the original Hamiltonian?). I guess this correspondence is also the motivation to consider the TM spectrum an observable at the correlation length scale, or is there another argument for that?

The TM, is an object that has already attracted a lot of attention, since in the translationally invariant setup, is what dictates all correlations in space of the state. Here, furthermore, given that we are interested in the continuum limit of a relativistic invariant lattice model, such correlations are the ones that determine also the spectrum of the field theory in the continuum. As a result, they are the fingerprint of the field theory. Again, this is based on the emerging low-energy relativistic invariance, that given the linearity of the dispersion relation, ensures that correlations in space map to correlations in time up to a velocity factor. We guess this is exactly what the Referee has in mind in their question, and we hope that our added material at the beginning of Section 2.1 will further clarify this point.

The first part of Section 3 is quite nice. I am somewhat confused, nevertheless, by the notion of effective Hamiltonian after Eq. (14). Is it a Hamiltonian whose ground state is the finite b. d. MPS, as opposed to the actual ground state of (12).?

Yes, this is the idea. Given an MPS, it always has a parent Hamiltonian, of which it is the exact ground state. In general such Hamilton acts on several sites, but given that we are interested in the continuum limit we can study it from first principles. In other words, the effective Hamiltonian defines the effective field theory (EFT) which is only correct up to scales of the order of the correlation length and thus this length define the characteristic size of the effective system. In the case of the equation in question, the system is of finite size, and the characteristic length scale is the system size $L$.

What is the relationship between \hat{H} and \hat{H}^*?

The Hamiltonian $\hat{H}(L)$ is precisely the one describing the finite-size system while $\hat{H}^*$ describes the fixed point Hamiltonian of the renormalization group, that acts on an infinite system at the critical point, which is, by definition, the only point where scale invariance is actually realized for all scales.

I also wonder why the choice was made to use the non-symmetric tensors in this section (leading to a perturbation by sigma) when symmetric ones are used in the following section (leading to a perturbation by epsilon), although I actually don't mind it because that way we get to see both cases.

As implicit in the Referee comment, this is one of the possible choices. In particular, it is the one that gives access to the most relevant operator, and thus the one which is easier to characterize, since it deviates from the fixed point faster than any other choice. By using generic, non-preserving $\mathbb{Z}_2$ tensors, we have access to the relevant operator with the lowest scaling dimension, in this case $x_\sigma = 1/8$, as opposed to $x_\varepsilon = 1$, which would be the only accessible in a symmetric scenario. We hence require lower system sizes to observe the relevant effects of this perturbation, which is of practical advantage. Later, we also consider the explicit preservation of the $\mathbb{Z}_2$ symmetry for completeness’s sake.

Right at the beginning of Section 5, what does it mean that "this spectral ratio is beyond perturbative field theory"?

The approximative schemes of perturbation theory are always based on power series in terms of a small quantity, most often a physical coupling constant, and describe the full interaction of the many-body physics in terms of orders of interactions that are interpreted within the respective perturbative picture. These power series have some radius of convergence for which the perturbative scheme is applicable, and fail beyond that. The data of Fig 2, which is what "this spectral ratio" is referring to, is generated through iDMRG simulations on an iMPS. DMRG simulations do not necessarily access a perturbative regime, and can contain arbitrarily non-perturbative effects.

By looking at the right panel of Fig. 3 we can see that the TM spectrum for finite size system starts perturbatively deviating from its expected behaviour as the finite bond dimension effects start to become relevant, and then it settles into a completely different scale invariant spectrum, which is only dictated by the bond dimension, and it is not perturbatively connected to the original spectrum. This is why we say that the effects of finite bond dimension in an MPS, cannot be interpreted within a perturbative picture unless the bond dimension is increased polynomially with the system size.

As a result, we need to introduce a new effective lattice model explaining the finite bond dimension effects of MPS at a critical point.

In this Section [5], I understand the second argument given to set m=1 (namely that it matches the numerics that we want the model to reproduce), but the first one (33)-(35) is a bit cryptic, I do not know what assumptions are being made in order to be able to use those formulas.

A: We acknowledge the perhaps confusing way we expressed ourselves. We have expanded a bit on the explanation and the discrepancy between the correlation length and the single particle gap, which are inversely proportional by definition. However, the proportionality factor does not need be trivial, and it is this factor which makes up the entire argument for setting $m=1$ in the case of the Ising model (as opposed to $m=-1$ for example). We have included a Reference to analytical forms of the correlation functions and made the argument clearer that $\xi = |\delta|^{-1}$ in the paramagnetic case ($g>1$) and $\xi = \frac{1}{2} |\delta|^{-1}$ in the SSB phase ($g<1$).

Similarly, for the argument before 5.2. that says that the same correlation length "costs" twice the energy when produced in the direction of the paramagnet.

We have expanded the explanations further on why the DMRG simulations must introduce a finite correlation length, which is equivalent to a perturbation as per Section 3, and why this perturbation must always settle in the SSB side of the phase transition when considering the transverse-field Ising model.

Also, was it checked somewhere that the correlation length of the ground state of H(L) goes as L (i.e. that nu is 1 under Figure 8) or is it an assumption?

Figures 5 and 6 corroborate the standard critical exponent $\nu=1$ in our lattice model, i.e., the scaling behavior of the correlation length with the coupling $\delta$. There, we observe the first non-degenerate gap $\Delta_1 = 2|1-g|$ at exactly $g = 1 - 1/L$ for any $L$ and in the thermodynamic limit. This implies $\xi \sim 1/\Delta \sim \delta^{-\nu}$ with $\nu=1$, and $\delta = 1/L$ which is precisely the scaling of the perturbation $\delta$ such that the transverse field operator is rendered marginal in combination with its coupling.

Finally, in general I find the notation and the reasoning in Appendix A a bit confusing. We assume that there is a perturbation that generates a correlation length, and then we incur in an error from truncating to finite bond dimension, but if the perturbation that induces the correlation length is the finite bond dimension itself, then there should not be an error due to truncation since psi_0 is already the ground state of the perturbed system, with the finite bond dimension in place, should it?

In this appendix, we only recite the work from Refs. [15] and [16]. The main idea is to assume the existence of a finite correlation length. This by itself implies that the energy is different from the ground state energy that would require an infinite correlation length, call it $\delta E(\xi)$. On the other hand one can take the critical ground state and truncate its Schmidt spectrum to a given bond dimension. This approximation also produce an error in the ground state energy, call it $\delta E(D)$. By asking which bond dimension would induce the same error as the one induced by the finite correlation length, namely asking $\delta E(\xi)= \delta E(D)$ then one arrives to an expression of the correlation length as a function of the bond dimension.

More details can be found in the original works.

  1. On page 3, the sentence "This has been the subject of several studies" is not followed by a reference.

A: We have taken out that sentence as we discuss these references in the paragraph just below.

  1. Beginning of page 5, is "ratio of the gap ratio" the correct expression?

A: Certainly not, thank you for pointing out the typo.

  1. For the unfamiliar reader, it could help to refresh what the scaling hypothesis is. E.g. page 6, "by virtue of the scaling hypothesis, all universal quantities only depend on the correlation length". Is it all there is to the scaling hypothesis, or is it a consequence of a more fundamental assumption?

This is indeed a concise recitation of the scaling hypothesis. We have added a clarifying sentence and a reference.

  1. In Fig. 2, the non-symmetry enforcing case is called symmetry broken instead. Is this intentional? Does the Hamiltonian actually have a degeneracy?

We have changed the label of Fig 2 to consistently call the non-symmetry preserving spectrum non-conserving. Indeed, the Ising model has a two-fold degenerate ground state in the ferromagnetic phase (g < 1), which is spontaneously broken. We have added a reference when introducing the spontaneous symmetry-breaking (SSB) acronym on page 18.

  1. The text after Eq (13) is a bit misleading: this equation represents the difference between the puMPS and the finite size energy, while the plot shows the difference between the puMPS value and the infinite size energy.

A: Thank you for pointing that out, we adapted the phrasing in the main text.

  1. On page 13, "transitionally" should read "translationally".

A: Thanks for spotting this malicious autocorrection typo.

  1. Typo in the caption of Fig 6, 1-1/xi instead of 1-1/L.

A: Thanks for spotting this typo.

  1. On page 21, "the theory is confined to the region", I think I know what is meant: in the spatial region where dS blows up, fluctuations of any field are suppressed and thus effectively log xi is a boundary, i.e. an interface with the vacuum. I would verbalize it a bit more, though.

A: We added the following clarifying sentence below E. (40): Since the perturbation $\hat{\Phi}_g$ is relevant, the RG dimension $(2-\Delta_g)$ is positive. Since the contribution to the partition function is weighted by $e^{-S}$ in the path-integral, the configuration with larger $\delta S$ has a smaller contribution.

  1. On page 22,"effectively halting it as the relevant operator was marginal Section 3" is missing an "if" and brackets?

A: Yes, this required some rephrasing. We have changed the wording of the sentence by adding a subclause specifying the marginal nature of the relevant operator when combining it with its scale dependent coupling

  1. Right before A.11, should ket(psi) be ket(psi_0) everywhere? Also e_0 and e seem to be the same from (A.6),

A: Indeed, to both! Thank you.

  1. In appendix B, the model is only KW invariant at g=1 (also the transformation would more accurately be represented as sigma_Z -> sigma_X sigma_X and vice versa). What is the value of p_c?

A: Correctly pointed out that the original Kramers--Wanier transformations do not apply here. We have corrected the duality transformation and copied the original ones from Alcaraz in Ref. [59]. Furthermore, Alcaraz tested the model within $0 \leq p \leq 1.5$ and showed that it is in the Ising universality class approximately until $p \approx 1.5$, cf. Fig 1 in Ref [59] and text page 5. We are unaware of a numerical study estimating $p_c$ more rigorously.

Attachment:

list-of-changes_Q7jkuUM.pdf

---

## Round 2 · Referee Report · Anonymous (Referee 2) · 2025-3-25

Report
I thank the authors for their careful, point-by-point reply to my report. I understand the work more clearly now and I support its publication. I attach a series of comments from my second reading of the manuscript as well as of the author's reply.
-
The explanation of why we are interested in the spectrum of the transfer matrix makes sense. I would make a bit more emphasis on the fact that the correspondence between the spectra of the TM and the effective Hamiltonian is a numerical observation (unless there is a proof of it in some reference). I would suggest also adding is a comment about the fact whether the transfer matrix is Hermitian, since it feels like it need not be, yet its spectrum is real anyway.
-
In Fig. 3(a), the reader is not given an explanation for the appearance of $\nu c$ in the cyan dotted line, the expansion of the $E_{\text{exact}}(L)$ only gives $\pi/6L$.
-
At the end of the first paragraph of page 14, it reads "Their distance from the unstable CFT fixed point increases logarithmically as we increase the system size." What distance measure are we talking about?
-
Above Eq. 25, I would use a different letter to count the number of fermionic modes, since m already means something else.
-
If I have now understood the Eqs. (33)-(35) argument correctly, the statement is that we can compute the correlation length exactly at both sides of the phase transition, thanks to (33) (which, I'd say, is a nontrivial result, as opposed to the author's reply saying that correlation length and single particle gap are inversely proportional "by definition"), and we pick the value of $m$ so that $\xi=L/2$. Is this the case?
-
On the sixth line of page 19, I would advise against using footnotes that can be mistook for exponents. Also, is there a reference for the information in the footnote (the bias of the variational energy) or is it an observation of the authors?
-
Towards the end of 5.2.2. there is a reference to Fig. 8(c) which does not exist.
-
I find the explanation for the influence of the choice of symmetric vs. nonsymmetric tensors on the spectrum of the TM is a bit rushed and sandwiched in the discussion of a different spectrum (the entanglement Hamiltonian spectrum, last paragraphs before Section 6), making it easy to miss. It would be nice to give it some more entity and discuss whether the relevant boundary conditions can be implemented in the toy model of the paper.
-
I appreciate the explanation about Appendix A, although I believe it does not match what is done in the Appendix. Having checked the references, it seems like Appendix A uses the argument of Pollmann et al., while the reply to my report features the argument of Pirvu et al. I think I understand now that the argument in the Appendix is to find the state, among the family of states with all possible correlation lengths, that minimizes the energy upon truncation to a given bond dimension w.r.t. the gapless Hamiltonian, and my confusion is why do we assume that the Schmidt values follow the distribution of a critical state also for states with finite correlation length, i.e., why (A.7) should follow (A.1), and why do we seem to use the gap of the perturbed Hamiltonian in (A.11) in order to determine the energy with respect to the gapless Hamiltonian. In any case, since this is based on previous published works, I guess it does not really affect this work that much.
-
Regarding my comment about Appendix B, I would say the transformation is indeed the original Kramers-Wannier. I guess the confusion was just that I was thinking self-duality as the Hamiltonian mapping back to itself under the duality (which only holds when $g=1$), while the authors are referring to the Hamiltonian mapping back to one in the same family (i.e. self-duality of the entire class).
Recommendation
Publish (easily meets expectations and criteria for this Journal; among top 50%)
Author: Jan Thorben Schneider on 2025-04-07 [id 5349]
(in reply to Report 1 on 2025-03-25)We thank the referee for their positive evaluation of our previous feedback and for the careful critique of our revised second version. Below, we intend to respond satisfactorily to all points raised by the referee.
A: We have added the clarification of the numerical observation of the mapping of the TM spectrum to the effective Hamiltonian spectrum. On the second point, indeed the TM spectrum does not need to be real as the transfer matrix is not Hermitian in general. In the Ising model, everything is real-valued because the Ising model has not just a Hermitian Hamiltonian but indeed a symmetric real Hamiltonian. The ground state of the Ising model is thus real-valued, and therefore so is the transfer matrix. We added a note stating that this is generally not true.
A: On page 11, in the main text discussing Fig. 3(a), we cite Refs. [53, 54], the relevant references which give the definition of the Casimir energy in the context of a CFT given the features of the model, i.e., speed of sound $v=2$, and conformal charge $c=1/2$. Note that the referee mistakenly read the cursive Latin letter $v$ as the cursive Greek letter $\nu$, possibly mistaking it for the critical exponent $\nu=1$. We have added a footnote to warn the reader. We have made the formula explicit in the main text in the new version, explaining the contributing factors in the Ising model. Essentially, $v \cdot c = 1$ as $v=2$ and $c=1/2$ in the Ising model, which hopefully should resolve the confusion.
The logarithmic distance originates from the RG flow equations which are always Ordinary Differential Equations (ODEs) in the logarithm of the system size, as in Eq. (15). Thus, the "flow" in the RG sense happens in a space of couplings with a notion of distance (or time if you want) given by the logarithm of the ratios of two system sizes, $\log(L_2/L_1)$, which are given by the initial and final point of the flow, $L_1$ and $L_2$, respectively. The maximum distance RG can flow in a physical system with system size $L$ is $\log(L/1)$, that is, from the microscopic definition of a single site (or unit cell), $L_1=1$, up to the notion of the complete system, $L_2=L$. The distance of the initial coupling flowing due to RG to its final point hence grows logarithmically with the size of the system and approaches the fixed point in the thermodynamic limit.
A: Thank you for noticing the double notation. We have adapted the label for the number of fermion modes.
A: We have restructured the arguments slightly and taken out the previous Eq. (33), which only served to give the speed of sound (or speed of light) $v$ the entrance as a necessary proportionality constant into the well-known inverse proportionality relation of the correlation length to the gap, for dimensional reasons. In addition to the dimensional argument, i.e., $\xi \propto v |\Delta_1|^{-1}$, there is a dimensionless factor of 2 arising from the domain-wall nature of the quasi-particle excitations. Furthermore, we have argued above Eq. (23), the definition of the lattice model, that we expect a factor of 2 between the correlation length and the system size, since the correlation of the effective Hamiltonian is mediated through the finite bond dimension tensors on both sides. As the induced spatial dimension of the finite bond dimension is given by the correlation length, this gives rise to a notion of the total system size given by twice the correlation length, $L = 2\xi$. We furthermore numerically verify our thusly constructed model, giving rise to the correct behavior. This consequently fixes $|m| = 1$ according to Eq. (37). The only ambiguity left is the sign of $m$, which is equivalent to which side of the phase transition does $\delta$ settle, i.e., in the SSB or the PM phase. We show that statement with the same arguments as before, which also prompted us to include the graphic showing the approximate symmetry of the variational ground state error w.r.t. the coupling $\delta$. This additionally yields the reference to the claim in the footnote the referee was asking for in the next question just below. We will resubmit this manuscript with the reference to this comment and its attachment and acknowledge Referee 2 for their constructive, clarifying, and helpful critique.
A: Thank you for suggesting this clearer typographic choice. We have swapped the footnote marker to symbols. We do not have a direct source for the bias of the variational energy. We computed it numerically exactly through the free fermion framework and the numerical packages of Ref. [64]. We have attached a plot to this comment, which shows the variational ground state energy error as a function of the perturbative coupling $\delta$. We hope this plot clears things up.
A: Thank you for spotting that typo. The label is now corrected to Fig. 8(a).
A: We appreciate the remark and acknowledge the brevity with which we go over this specialized detail. We have expanded the explanations and added a discussion of the matter in Section 2 and 3. Furthermore, at the very end of Section 2, when we previously talked about shedding more light onto the puzzle of the different spectra of the TM when enforcing the $\mathbb{Z}_2$-symmetry or not, we have added the cross-reference to the section (Section 5.2.3) where we discuss this in more detail.
A: On the implementation of the boundary conditions in the toy model of our paper: Indeed, it remains open to further investigations to incorporate other boundary conditions in our toy model which could correspond to different spectra.
Attachment:
MPS-TFM-attachement-plot.pdf

---

## Round 2 · Author Response

List of changes
- Page 3, removed the sentence “This has been the subject of several studies.”
- Beginning of Section 2: We have made the distinction of thermodynamic limit and
infinite degrees of freedom
- Page 6, added sentence on further details on the scaling hypothesis and reference.
- Page 5, thus affecting ``thus affecting the ratio of the gap ratios'' to ``thus affecting the ratio of the gaps''
- Beginning of Section 2.1: Added a paragraph on introduction on the transfer matrix and motivation of studying MPS TMs.
- Fig 2 description: symmetry-broken to non-conserving
- Figure 3(b): x-axis label changed from $L/\xi_0 \chi^\kappa$ to $L/\xi(\chi)$, label for dashed green line changed from $L^{15/4}$ to $(L/\xi)^{15/4}$
- Below Eq.~(13) and describing Fig.~3(a) correction of quantity explained from ``this relation is shown'' (i.e. $\delta E_0$) to ``$E_0 - \epsilon \, L$'' and in the sentence referencing Fig.~3(a), a clarification and definition of $\epsilon_0$
- Page 12. This result aligns with our numerical observations shown in Fig.~3(b) [added:] when replacing the renormalization scale \(L\) with \(L/\xi(\chi)\), being the relevant length scale in the crossover regime from finite-size to finite-entanglement scaling.
- Below Eq.~(34) add spontaneous:
However, in the spontaneous symmetry-breaking phase (SSB), the quasi-particle excitations are domain walls. Add also sentence and reference to algebraic Bethe ansatz with its analytical solutions.
- On page 13, changed ``transitionally'' into ``translationally''.
- Typo in the caption of Fig 6, $1-1/\xi$ instead of $1-1/L$.
- Page 21, added clarification below Eq. (40) on the confinement of the theory
- Page 17, Eqs. (34) and (35), added expression in terms of the perturbation $\delta$.
- Page 17, define acronym SSB as ``spontaneous symmetry-breaking phase''
- Page 18, adding further clarification for why the SSB phase is selected by iDMRG.
- On page 22, added subclause specifying that the relevant operator is rendered marginal in combination with its scale dependent coupling (twice)
- Right before A.11, corrected $\ket{\psi}$ to be $\ket{\psi_0}$, and $e$ to $e_0$.
- Below (B.1) updated the self-duality mapping to the correct one from the original reference.
- Below Eq.~(34): moved the sentence from before Eq.~(35) about the two elementary excitations needed to create any correlation
- Several minor orthographic corrections

---

## Round 2 · List of Changes

- Page 3, removed the sentence “This has been the subject of several studies.”
- Beginning of Section 2: We have made the distinction of thermodynamic limit and
infinite degrees of freedom
- Page 6, added sentence on further details on the scaling hypothesis and reference.
- Page 5, thus affecting ``thus affecting the ratio of the gap ratios'' to ``thus affecting the ratio of the gaps''
- Beginning of Section 2.1: Added a paragraph on introduction on the transfer matrix and motivation of studying MPS TMs.
- Fig 2 description: symmetry-broken to non-conserving
- Figure 3(b): x-axis label changed from $L/\xi_0 \chi^\kappa$ to $L/\xi(\chi)$, label for dashed green line changed from $L^{15/4}$ to $(L/\xi)^{15/4}$
- Below Eq.~(13) and describing Fig.~3(a) correction of quantity explained from ``this relation is shown'' (i.e. $\delta E_0$) to ``$E_0 - \epsilon \, L$'' and in the sentence referencing Fig.~3(a), a clarification and definition of $\epsilon_0$
- Page 12. This result aligns with our numerical observations shown in Fig.~3(b) [added:] when replacing the renormalization scale \(L\) with \(L/\xi(\chi)\), being the relevant length scale in the crossover regime from finite-size to finite-entanglement scaling.
- Below Eq.~(34) add spontaneous:
However, in the spontaneous symmetry-breaking phase (SSB), the quasi-particle excitations are domain walls. Add also sentence and reference to algebraic Bethe ansatz with its analytical solutions.
- On page 13, changed ``transitionally'' into ``translationally''.
- Typo in the caption of Fig 6, $1-1/\xi$ instead of $1-1/L$.
- Page 21, added clarification below Eq. (40) on the confinement of the theory
- Page 17, Eqs. (34) and (35), added expression in terms of the perturbation $\delta$.
- Page 17, define acronym SSB as ``spontaneous symmetry-breaking phase''
- Page 18, adding further clarification for why the SSB phase is selected by iDMRG.
- On page 22, added subclause specifying that the relevant operator is rendered marginal in combination with its scale dependent coupling (twice)
- Right before A.11, corrected $\ket{\psi}$ to be $\ket{\psi_0}$, and $e$ to $e_0$.
- Below (B.1) updated the self-duality mapping to the correct one from the original reference.
- Below Eq.~(34): moved the sentence from before Eq.~(35) about the two elementary excitations needed to create any correlation
- Several minor orthographic corrections

---

## Round 3 · List of Changes

Below is a list of changes from v2 to v3 to the preprint with arXiv identifier \texttt{arXiv:2411.03954}.

  • Page 8, added 'numerically' to 'observed numerically' in the first paragraph of 2.1.

  • Page 9, expanded the explanation on the symmetry-perserving simulation and added reference to later Section.

  • Page 9, added clarification that the TM spectrum in the Ising models is real, but not generically.

  • Page 11, expanded definition of the Casimir energy contribution and explained factors.

  • Page 12, added a paragraph on the effect of renormalization on the symmetry-preserving and non-preserving simulations of the Ising model.

  • Page 16, above Eq. (25), changed $m$ to $l$ when counting the number of fermionic modes.

  • Page 17, 18, we have taken out previous Eq. (33), the single particle gap. We have also slightly restructured the chain of arguments setting $m=+1$ in our lattice model definition. Lastly, we added a reference to the reply to the report of Referee 2, which includes an attached file showing the plot that verifies that the variational ground state energy error is approximately symmetric w.r.t. the coupling $\delta$

  • Page 18, corrected the wrong placement of the factor $1/2$ in previous Eqs. (34) and (35)

  • Page 18, moved the paragraph explaining why the iDMRG algorithm always chooses the SSB phase as opposed to the PM phase to this spot. Previously, it was located just before Sec. 5.2.

  • Page 20, corrected label Fig. 8(c) to Fig 8.(a).

  • Several minor orthographic corrections

---

## Editorial Decision

published